# Fragments to Facts: Partial-Information Fragment Inference from LLMs

Lucas Rosenblatt [* 1]   Bin Han [* 2]   Robert Wolfe [2]   Bill Howe [2]

## Abstract

Large language models (LLMs) can leak sensitive training data through memorization and membership inference attacks. Prior work has primarily focused on strong adversarial assumptions, including attacker access to entire samples or long, ordered prefixes, leaving open the question of how vulnerable LLMs are when adversaries have only partial, unordered sample information. For example, if an attacker knows a patient has "hypertension," under what conditions can they query a model fine-tuned on patient data to learn the patient also has "osteoarthritis?" In this paper, we introduce a more general threat model under this weaker assumption and show that fine-tuned LLMs are susceptible to these fragment-specific extraction attacks. To systematically investigate these attacks, we propose two data-blind methods: (1) a likelihood ratio attack inspired by methods from membership inference, and (2) a novel approach, PRISM, which regularizes the ratio by leveraging an external prior. Using examples from medical and legal settings, we show that both methods are competitive with a data-aware baseline classifier that assumes access to labeled in-distribution data, underscoring their robustness.

## 1. Introduction

Instruction-tuned language models (LLMs) have transformed how users interact with AI, offering personalized assistance in domains ranging from fact-checking to postpartum care (Tang et al., 2024; Antoniak et al., 2024; Wolfe & Mitra, 2024a;b). This tight feedback loop between end users and developers also enables continuous fine-tuning of LLMs on the very data collected from user interactions (Poddar et al., 2024). Yet, as these models grow in size and scope,

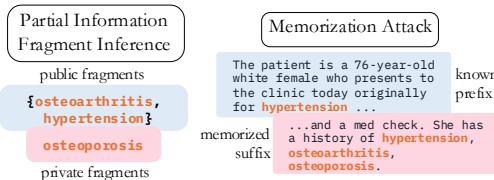

Figure 1. Comparing the PIFI LLM threat model to the memorization threat model in a medical scenario. PIFI uses unordered, publicly available *fragments* from a sample (like a patient record) to infer private fragments (like a sensitive medical condition). Memorization assumes access to an *ordered* prefix of the sample, and checks for verbatim generation by an LLM of the suffix.

they've become susceptible to attacks targeting private, sensitive information (Carlini et al., 2021; 2024).

A growing body of work has studied *membership inference attacks* which aim to determine if a specific example appears in the training set (Jagannatha et al., 2021; Carlini et al., 2022a; Mireshghallah et al., 2022; Shi et al., 2023; Mattern et al., 2023; Morris et al., 2023; Duan et al., 2024) and *memorization attacks* which attempt to reconstruct verbatim training examples (Carlini et al., 2021; 2022b; 2024; Lukas et al., 2023). Membership inference often targets outlier samples, while memorization attacks focus on content that is repeated ("inlier" samples (Dionysiou & Athanasopoulos, 2023)), and each approach typically assumes strong adversarial knowledge: at least access to ordered prefixes, and typically access to entire training samples (for example, in the case of copyright infringement cases (Grynbaum & Mac, 2023; Small, 2023)). These assumptions overlook more realistic and common scenarios in which an attacker possesses only *partial, unordered* information about a target.

We consider a malicious adversary who seeks to acquire sensitive *fragments of information* from an individual's data. For example, consider an attacker who learns that an individual has hypertension from a social media post, and wants to learn co-morbid conditions that apply to that individual in order to target sales, deny insurance, or even aid in identity theft. Using this small subset of the patient's data (*i.e.*, a single diagnosed condition), can the attacker extract those co-morbidities by prompting a fine-tuned language model? Though they remain under-specified and under-studied, such attacks are increasingly plausible, especially in the case of a clinic or hospital offering a public or internal chatbot created by fine-tuning on the organization's own medical data

*Equal contribution [1]New York University, New York, USA [2]University of Washington, Seattle, USA. Correspondence to: Lucas Rosenblatt <lr2872@nyu.edu>.

*Proceedings of the 42nd International Conference on Machine Learning*, Vancouver, Canada. PMLR 267, 2025. Copyright 2025 by the author(s).

(McDuff et al., 2023; Wu et al.; Zhou et al., 2024).

In this research, we propose a new *partial-information fragment inference* threat model that unifies insights from membership inference and extractable memorization attack strategies under weaker adversarial assumptions. Notably, we show that if an attacker only knows a small set of text fragments from an original training sample (*e.g.*, that "hypertension" and "beta blockers" are in a patient's medical notes), they can infer additional, potentially sensitive fragments (*e.g.*, "osteoporosis") that are also in the training sample.

**Contributions** Our findings reveal that even weak adversaries with access to only a few unordered fragments of an individual's sample can pose a privacy threat by comparing results between models to infer additional fragments. Based on our results, we suggest the need for improved defenses that mitigate not just sample memorization or membership inference, but also partial, fragment-level inference vulnerabilities, **before** deployment of public-facing fine-tuned models in sensitive domains like medicine or law. More specifically, we make the following contributions:

**①  A Novel Threat Model.** We formalize a threat model where an adversary only has access only to publicly available *text fragments* for a target individual, as opposed to that individual's full sample.

**②  Effective Data-Blind Attacks.** We show that LR-Attack, a straightforward **L**ikelihood **R**atio approach, is surprisingly competitive, often rivaling a more powerful Classifier baseline that assumes access to labeled, in-distribution examples. We also propose PRISM (**P**osterior-**R**efined **I**nference for **S**ubset **M**embership), a method that uses the LR-Attack score to update a posterior likelihood by using a prior, ultimately reducing false positives.

**③  Empirical Validation.** We conduct experiments in a *medical summarization* context, first fine-tuning an LLM for summarization on medical notes. We then reduce each note to a set of fragments (in this case, medical terms), and we simulate the efficacy of our attacks in the hands of an adversary who possesses only a small amount of information about an individual. Our experiments show that fine-tuned LLMs are vulnerable to extraction attacks under even these limited-information conditions; we observe a *9.5%* TPR on Qwen-2-7B at 2% FPR using LR-Attack, and an *11.5%* TPR on Llama-3-8B at 5% FPR using PRISM, for example.

We describe the problem setting and notation in §2, present our threat model in §3, attack approaches in §4, and describe experimental setup on real-world scenarios in §5. We introduce the LLMs in §6, and articulate experimental results in §7. We then discuss related work, implications, and challenges from our work, encouraging further exploration of strengths and limitations of this new family of attacks.

## 2. Preliminaries

We provide relevant background on LLMs and data privacy attacks, and we introduce necessary notation. We then discuss related adversarial threat models: sample membership inference and training data extraction.

**Large Language Models** We study generative (also known as "autoregressive" or "causal") language models, which are trained to predict the next word in a sequence by maximizing the negative log-likelihood of the model over tokens (words or subwords) in a vocabulary (Radford, 2018). We specifically examine generative language models that have undergone the process of instruction tuning (Wei et al., 2021), meaning our models use a chat-based format wherein the model and user take turns exchanging messages, and the model responds in accordance with human preferences (Ouyang et al., 2022). Examples of such models include ChatGPT (OpenAI, 2022), Llama (Dubey et al., 2024; Touvron et al., 2023), and Mistral (Jiang et al., 2023) — models that are particularly relevant because of their ease of use and accessibility both to lay users and potential adversaries.

**Notation** We adopt several typical notational conventions when discussing language models and extraction attacks. Consider a token sequence $(x_1, x_2, \ldots, x_T) = \mathbf{x} \in \mathcal{X}^T$, where $\mathcal{X}^T$ is the space of possible input sequences of length $T$. Each $x_i$ is drawn from a set of vocabulary tokens $\mathcal{V}$ (*i.e.*, $x_i$ is *typically* a single word or word subsequence in $\mathcal{V}$). We refer to the dataset of token sequence samples as $D = \{\mathbf{x}_i\}_{i=1}^n$; we will also let $D \sim \mathcal{D}$, where $\mathcal{D}$ is a probability distribution over $\mathcal{X}^T$ specific to some task (*e.g.*, summarization of medical notes).

Formally, a generative language model is fit by learning to predict the next token over $\mathcal{X}^T$ by expanding,

$$\Pr(x_1, x_2, \ldots, x_T) = \prod_{i=1}^{T} \Pr(x_i \mid x_1, \ldots, x_{i-1}) .$$

We refer to a generative language model fine-tuned on a dataset as $f_{\theta, D}$ (read as model $f$ with pre-trained parameters $\theta$, further fine-tuned on $D$). We then denote the probability of a sequence of tokens under a given model as $f_{\theta, D}(\mathbf{x}) = f_{\theta, D}(x_n \mid x_1, \ldots, x_{n-1})$. To denote the probability of a sequence of tokens under a specific prompt sequence $\mathbf{x}$, we write $f_{\theta, D}(\mathbf{y} \mid \mathbf{x})$ or $f_{\theta, D}(y_1, \ldots, y_n \mid x_1, \ldots, x_m)$.

**Threat Model Specific Notation**

We also adopt several non-standard notational conventions. To help distinguish between an arbitrary training sequence and a sequence associated with an individual, we use the shorthand $(s_1, \ldots, s_n) = \mathbf{s}$ to refer to a sequence associated with a *single* individual in $D$ (*e.g.*, $\mathbf{s}$ refers to John Doe's medical note). We refer to short, ordered sequences of tokens as ***fragments***. Note that a *fragment*

can be a single *word* (*e.g.*, osteoporosis), yet consist of an *ordered sequence of tokens* in an LLM embedding matrix (*e.g.*, 'oste,' 'opor,' 'osis'). Additionally,

**1** We let the function $\mathcal{A}$ represent our *adversarial public information assumption*, which maps an ordered sample sequence $\mathbf{s} = (s_1, \ldots, s_n)$ to a potentially unordered set $\mathcal{S}$ of short *fragments*, typically keywords that carry sensitive information but are discoverable through public information. For example, an attacker might learn from social media that John Doe contracted covid and experiences hypertension. While the fragments "covid" and "hypertension" are a subset of John Doe's complete medical chart, no prefix or other significant component of the chart itself is available. Formally, let $\mathcal{A} : \mathcal{X}^n \to \mathcal{P}(\mathcal{X}^*)$, so that $\mathcal{S} = \mathcal{A}(\mathbf{s})$. Usually we assume $\mathcal{S} \subset \mathrm{Frag}(\mathbf{s})$, where $\mathrm{Frag}(\mathbf{s})$ denotes the collection of all fragments contained in $\mathbf{s}$, but $|\mathcal{S}| \ll |\mathrm{Frag}(\mathbf{s})|$ and the fragments in $\mathcal{S}$ are words or short phrases.

**2** We let $\mathbf{y}^*$ denote a single, private *candidate fragment* targeted under our threat model.

**3** We use the conventional term *shadow model* (Carlini et al., 2022a) to discuss a model fine-tuned on data from distribution $\mathcal{D}$, but where we are sure that the sample of interest $\mathbf{s}$ was *not* included. Formally, let $D'$ denote a shadow dataset, where $\mathbf{s} \notin D'$; both $D$ and $D'$ are drawn from $\mathcal{D}$. Thus, finetuning on $D'$ yields a *shadow model*, $f_{\theta,D'}$. Finally, we introduce notation for a *world* model that allows us to query $k$ arbitrary models to estimate the probability of a sequence $\mathbf{x}$ under any model. We thus denote the world model $f_{\theta,\mathtt{world}}$ as follows: $f_{\theta,\mathtt{world}}(\mathbf{x}) = \frac{1}{k} \sum_{i=1}^{k} f_{\theta,*}^{(k)}(\mathbf{x})$.

## 2.1. Prior Threat Models

We outline prior work that induces language models to expose specific, private information learned from training data.

**Standard Language Model Attacks** Two prominent attack threat models have emerged in the literature on LLM privacy. The first, extractable memorization (*EM*) (Carlini et al., 2019; 2021; 2022b), (sometimes called *discoverable* memorization (Wang et al., 2024)), asks whether prompting a model with part of a sample from its training data will lead the model to output the rest of the sample (approximately (Yu et al., 2023) or exactly (Lukas et al., 2023)). The second, *membership inference* (*MI*) (Carlini et al., 2022a), asks whether one can infer sample membership in the training data from model outputs (usually as a binary classification task). We describe both attacks to motivate our approach.

Carlini et al. (2022b) define *extractable memorization* as follows: for a black-box target model $f_{\theta,D}$, a sample suffix $\mathbf{s}_{k:n}$ is considered *extractable* if there exists a length-$(k-1)$ prefix, $\mathbf{s}_{1:k-1}$, such that the concatenation $[\mathbf{s}_{1:k-1} \| \mathbf{s}_{k:n}]$ is in $D$ and $f_{\theta,D}$ produces $\mathbf{s}_{k:n}$ when decoded greedily under prompt $\mathbf{s}_{1:k-1}$. Conversely, given black-box access to a target model $f_{\theta,D}$ and a complete target sequence $\mathbf{s}$, a *membership inference* attack produces a likelihood that $\mathbf{s}$ was in $\mathcal{D}$, mediated by the output of $f_{\theta,D}$ (Carlini et al., 2022a). *MI* attack performance is based on using that likelihood to make a classification (*i.e.*, predicting 1 or 0 for included or not included, respectively). Despite differing objectives, the threat models share an assumption: they assume *a priori* access to $\mathbf{s}$, a *complete and ordered* target sample of interest.

# 3. Partial-Information Fragment Inference

Assuming *a priori* access to a complete sample $\mathbf{s}$ is plausible in some settings, like measuring the likelihood of copyright infringement of *one's own data*, but not in cases where an adversary attacks a model without access to the original training data. We thus propose the *partial-information fragment inference* (PIFI, pronounced "piffy") adversarial threat model for information extraction under a weaker assumption; namely, under PIFI, an adversary seeks to infer specific and sensitive fragment-level information about individuals based on a *small subset of public fragments* (*e.g.*, between 4 and 30) assumed to be present in sample $\mathbf{s}$.

## 3.1. Capabilities of the Adversary

The PIFI threat model makes the same black-box model access assumptions as most *MI* and *EM* attacks. Specifically, PIFI assumes access to output probabilities produced by $f_{\theta,D}$, $f_{\theta,D'}$, and arbitrary additional models $f_{\theta,\mathtt{world}}$, but not access to model weights, embeddings, or training parameters. Unlike other threat models, PIFI does *not* assume access to a complete sample $\mathbf{s}$ associated with an individual; instead, the adversary only has access to a fragment set $\mathcal{S} \subset \mathrm{Frag}(\mathbf{s})$. The adversary can then *investigate* a candidate target fragment $\mathbf{y}^*$, where $\mathcal{S}$ is useful for investigating $\mathbf{y}^*$ if $\mathcal{S}$ is likely to be a subset of the tokens in the original sample. Consider the realistic scenario wherein an attacker has learned something specific about an individual's medical history from their social media: PIFI enables the attacker to use that limited information as $\mathcal{S}$ to infer additional sensitive data $\mathbf{y}^*$ about the individual from an LLM $f_{\theta,D}$.

## 3.2. Goal of the Adversary

The adversary would like to know if the candidate target fragment $\mathbf{y}^*$ is in an individual's sample $\mathbf{s}$, given that $\mathbf{s}$ was included in training data $D$. Why is it important to condition on an individual's sample $\mathbf{s}$ actually being included in $D$? If the adversary had reason to believe that $\mathbf{s}$ was *not* in $D$, then

*Figure 2.* An illustration of our threat model. An LLM is fine-tuned, *e.g.*, with private medical notes. Then, an adversary prompts the fine-tuned LLM with relevant *fragments* of information (*e.g.*, from a target individual's medical records) to infer unknown fragments associated with the individual.

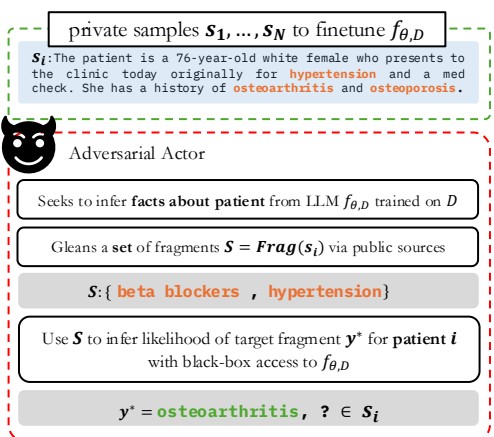

their inference might be about individuals with samples *like* **s**, and not about the *specific* individual associated with **s**. For example, any individual with *osteoarthritis* may also have increased odds of being a *smoker*; however, this does not mean that Jane Doe, who has *osteoarthritis*, is also a *smoker*, unless we have reason to believe that her sample **s** (*e.g.,* her medical note) says this is the case.

---

**Algorithm 1** A Class of PIFI Attack Models

**Input:** Private fragment $\mathbf{y}^*$, public fragment set $\mathcal{A}(\mathbf{s}) = \mathcal{S}$ for an individual, target language model $f_{\theta,D}$, shadow and world models $f_{\theta,D'}$, $f_{\theta,\text{world}}$, decision threshold $\tau$,

**Output:** {0,1}

1: $p_D = f_{\theta,D}(\mathbf{y}^* \mid \text{Prompt}(\mathcal{S}))$
2: $p_{D'} = f_{\theta,D'}(\mathbf{y}^* \mid \text{Prompt}(\mathcal{S}))$
3: $p_{\text{world}} = f_{\theta,\text{world}}(\mathbf{y}^* \mid \text{Prompt}(\mathcal{S}))$
4: $\ell \leftarrow \text{INFER}([\, p_D,\, p_{D'},\, p_{\text{world}} \,])$, where $\ell$ scores the likelihood that $\mathbf{y}^* \in \mathbf{s}$ given $\mathbf{s} \in D$.
5: **Return** $\mathbb{1}\left[\ell > \tau\right]$.

---

In Algorithm 1, we formalize the inputs and outputs for the PIFI threat model, and we constrain the class of INFER functions that we consider in this paper. Specifically, we consider adversaries that use their access to models $f_{\theta,D}, f_{\theta,D'}$ and $f_{\theta,\text{world}}$ to compute quantities like, $p_D = f_{\theta,D}(\mathbf{y}^* \mid \text{Prompt}(\mathcal{S}))$, where $\text{Prompt}(\mathcal{S}) = [\mathbf{x_i}, s_1, \ldots, s_m, \mathbf{x_j}]$ is a simple prompt in which the fragments possessed by the adversary are embedded. Accordingly, using only information available in the vector of probabilities $[p_D,\, p_{D'},\, p_{\text{world}}]$, the attacks PRISM, LR-Attack, and Classifier each specify an INFER function to produce a *likelihood* $\ell$ corresponding to the belief that $\mathbf{y}^*$ is in **s**, given that **s** is in $D$, which can be converted to a binary prediction with a decision threshold $\tau$.

## 4. Approaches to PIFI

In this section, we describe the class of INFER functions for producing the likelihood $\ell$. Then, we define the three specific functions from this class that we consider in this work: Classifier, LR-Attack, and PRISM.

### 4.1. The Classifier Baseline

To produce a strong baseline, we train a binary classifier on the same three-dimensional vector of probabilities $[p_D,\, p_{D'},\, p_{\text{world}}]$ used in the computation of our LR-Attack and PRISM attacks, with the goal being to predict whether a given $\mathbf{y}^*$ was in the data $D$ on which a target LLM was trained. For training, however, a classifier requires *labeled data*. Labeled data access is an unrealistic assumption under the PIFI threat model; the attacker explicitly does *not* have access to any amount of the training data $D$ for target model $f_{\theta,D}$, and may not even have access to samples suitably similar to target **s**. Of course, if the attacker somehow *did* have labeled training data, they could adjust their INFER approach using standard machine learning tools to fit the distribution over labels, leading to a strong attack. This makes Classifier, which we call a *data-aware* method, a very strong reference point for the performance of LR-Attack and PRISM, which are *data-blind* methods. Generally, we expect it to upper-bound their performance, assuming the training distribution generalizes for a particular distribution over text fragments. Note that for the empirical evaluations included in this paper, we employ a Light Gradient Boosting Machine ("LightGBM") model (Ke et al., 2017), a highly performant option for binary classification with data, and do standard cross-validation when training. Overall, Classifier is an important validation of the PIFI threat model and a target for the *data-blind* attacks; Classifier validates that there *is* signal in the vector $[p_D,\, p_{D'},\, p_{\text{world}}]$ for attacks under the PIFI threat model (assuming one can specify the right *data-blind* INFER function to exploit it).

### 4.2. The LR-Attack approach

The Neyman-Pearson Lemma is a classical insight from hypothesis-testing literature that generally underpins statistical hypothesis testing and likelihood based attacks, like the *data-blind* LR-Attack we propose (Neyman & Pearson, 1933). Intuitively, the Lemma states that, given two distributions, the *optimal* way to distinguish between them at a fixed false-positive rate is to threshold their *likelihood ratio*. Carlini et al. (2022a) reinterpret this in the context of membership inference: if $\mathbb{Q}_{\text{in}}(\mathbf{s})$ is the distribution over models trained *with* **s** and $\mathbb{Q}_{\text{out}}(\mathbf{s})$ is that over models trained *without* **s**, then the Neyman-Pearson Lemma implies we should estimate $\Lambda = \frac{p(f|\mathbb{Q}_{\text{in}}(\mathbf{s}))}{p(f|\mathbb{Q}_{\text{out}}(\mathbf{s}))}$, where $p(\cdot \mid \mathbb{Q}_{\text{in/out}}(\mathbf{s}))$ denotes the probability density of models under each distribution.

As Carlini et al. (2022a) note, we never have direct access to these true distributions, so we must approximate them empirically (*e.g.*, using shadow models).

Unlike the membership-inference threat model, however, PIFI does *not* assume access to the full sample **s**. Nonetheless, the Neyman-Pearson Lemma still suggests that comparing probabilities obtained from a target model potentially fine-tuned *with* **s** against those from a model fine-tuned *without* **s** can be informative. Concretely, for a candidate fragment $\mathbf{y}^*$ and known (public) fragment set $\mathcal{S}$, we measure $p_D = f_{\theta,D}(\mathbf{y}^* \mid \text{Prompt}(\mathcal{S}))$ and $p_{D'} = f_{\theta,D'}(\mathbf{y}^* \mid \text{Prompt}(\mathcal{S}))$. We then define $\hat{\ell} = p_D/p_{D'}$, which approximates the ideal likelihood-ratio statistic. Large $\hat{\ell}$ indicates that the target model assigns an unusually high probability to $\mathbf{y}^*$ relative to the shadow model, in the context of the known fragments $\mathcal{S}$. In the extreme case where we *already know* that $\mathbf{y}^*$ truly appears in **s**, distinguishing between $\{\mathbf{s} \in D\}$ and $\{\mathbf{s} \notin D\}$ reduces to membership inference — but in the PIFI setting, the goal is to infer whether $\mathbf{y}^*$ is in **s**, using only our *partial* knowledge $\mathcal{S}$ to guide that inference. Thus, while $\hat{\ell}$ does not directly compute the "true" membership-based ratio, we can expect it to sufficiently reflect that ratio to serve as an effective attack statistic.

### 4.3. The PRISM approach

Recall the example of Jane Doe in Section 3.2, where the attacker could mistakenly infer that Jane is a *smoker* simply because she has *osteoarthritis*. LR-Attack may suffer from such false positives: we cannot fully distinguish whether $\mathbf{y}^*$ is associated with $\mathcal{S}$ *specifically because* Jane's personal record is in $D$, or because individuals with $\mathcal{S}$ in their samples *generally* also include $\mathbf{y}^*$.

To remedy this, we consider how to incorporate an additional "world model" probability, $p_{\text{world}}$, into our likelihood score. We observe that one way to do this is to take a *prior* on sample membership, and try to derive the probability of interest. Suppose $D^* \sim \mathcal{D}$ is an unknown dataset used to train a language model, and let us write $\mathcal{S}$ short for $\text{Prompt}(\mathcal{S})$ here for clarity of presentation. Then, the probability that $f_{\theta,D^*}$ assigns to $\mathbf{y}^*$ given $\mathcal{S}$, *conditional* on $\mathbf{s} \in D^*$, can be expanded

$$\Pr(f_{\theta,D^*}(\mathbf{y}^* \mid \mathcal{S}) \mid \mathbf{s} \in D^*) =$$
$$\frac{\Pr(f_{\theta,D^*}(\mathbf{y}^* \mid \mathcal{S})) - \Pr(f_{\theta,D^*}(\mathbf{y}^* \mid \mathcal{S}) \mid \mathbf{s} \notin D^*)\Pr(\mathbf{s} \notin D^*)}{\Pr(\mathbf{s} \in D^*)}$$

If $D^* = D$, then $\Pr(f_{\theta,D^*}(\mathbf{y}^* \mid \mathcal{S}) \mid \mathbf{s} \notin D^*)$ can be approximated by $p_{D'}$. Similarly, we approximate $\Pr(f_{\theta,D^*}(\mathbf{y}^* \mid \mathcal{S}))$ with $p_{\text{world}}$, using queries to several "world" models that capture the *general* association between $\mathcal{S}$ and $\mathbf{y}^*$. It remains to estimate $\Pr(\mathbf{s} \in D^*)$. We connect this to our LR-Attack statistic, $\hat{\ell} = p_D/p_{D'}$. We assume $\mathcal{L}$ is strongly correlated with $\Pr(\mathbf{s} \in D)$ – an approximation

that may not hold but captures the intuition that if $\mathbf{y}^*$ is in **s**, then $p_D$ will be higher relative to $p_{D'}$. Hence, we choose to treat $\Pr(\mathbf{s} \in D)$ as a random event $M$ with some prior belief $\beta = \Pr(M)$, and then apply a function in the shape of a Bayesian update using the LR-Attack statistic, estimating $\hat{\Pr}(M \mid \hat{\ell}) := \frac{\Pr(\hat{\ell} \mid \mathbf{s} \in D)\ \Pr(M)}{\Pr(\hat{\ell})}$. We pragmatically assume $\Pr(\hat{\ell} \mid \mathbf{s} \in D) \propto \hat{\ell}$ and $\Pr(\hat{\ell} \mid \mathbf{s} \notin D) \propto 1/\hat{\ell}$. We plug $\hat{\ell}$ and $1/\hat{\ell}$ in directly to obtain a closed-form score we believe is correlated with the true $\Pr(M \mid \hat{\ell})$.

As we make many assumptions and approximations here, and it is difficult to reason about these distributions in the context of *large* language models, we investigated PRISM using a *small*, known language model under the PIFI threat model. In particular, we tested PRISM against LR-Attack on a simple trigram language model with a small, character-level vocabulary that we could fully specify (see Appendix C); this allowed us to *actually calculate* the above probabilities that we can only hope to approximate with larger language models. In this controlled setting, PRISM outperformed LR-Attack; this further motivated us to evaluate it alongside LR-Attack on LLMs.

## 5. Experimental Setup & Datasets

In this section, we describe the experimental setup and datasets for the empirical evaluation of our attacks in real-world scenarios (medical in §5.1; legal in §5.2). Code for this project is available at github.com/BeanHam/fragments-to-facts/.

### 5.1. Model Instantiation: Medical Summarization

Medicine is a highly sensitive domain, where language models are actively being trained on privately identifiable information (Das et al., 2024). Thus, we assess the efficacy of our attacks on a medical summarization task. We use the MTS-Dialog dataset (Abacha et al., 2023; Yim et al., 2023; Han et al., 2023), which includes 1,700 doctor-patient dialogues, with corresponding summaries. To construct fragment sets for our *adversarial public information assumption*, we run a biomedical Named Entity Recognition (NER) tool from SciSpacy[1] on each medical dialogue. Our fragments are then medically specific terms and phrases (*e.g.*, "constipation" or "acute cholecystitis"; see Appendix D for examples). We filter out dialogues without any extracted entities, leaving us 948 train, 69 validation, and 312 test samples. This process yields a variety of types of fragments for each sample, including medical conditions, drug names, and symptoms. Evaluating the PIFI threat model means *completely* disregarding the original dialogues and *focusing exclusively on*

---

[1]We use *en_ner_bc5cdr_md*, an NER model trained on the BC5CDR corpus, which contains annotated chemicals and diseases, https://allenai.github.io/scispacy/.

*the medical fragment sets* for inference.

We are careful to include a *variety* of false positive examples in our testing; see Figure 6 for an overview of attack failure modes. To construct negative examples, we sample a second disjoint set of *unseen* fragments, *i.e.*, fragments from notes that *did not appear in the fine-tuning subset* and which are not part of the samples that we test as true positives. The attacker must distinguish between true positive $\mathbf{y}^*$ fragments (*i.e.*, this $\mathbf{y}^*$ co-occurred alongside fragments from $\mathcal{S}$ in Prompt($\mathcal{S}$), given the $\mathbf{s}$ from which $\mathcal{S}$ was derived is in the training data $D$) and false positive fragments (anything else). Light experimentation led us to believe that the attack is not particularly sensitive to the choice of Prompt($\cdot$), so long as it describes the fine-tuning task (*i.e.*, says something about a *summary*). We thus formulate a consistent, task-appropriate Prompt($\cdot$) as follows: `Consider a patient whose medical summary contains: {diabetes, hypertension, ...}. That individual's medical summary also includes: ` $\{\mathbf{y}^*\}$. By inserting fragments, we obtain output probabilities from each model which serve as inputs for our `LR-Attack` and `PRISM` attacks, as well as our `Classifier` baseline.

### 5.2. Model Instantiation: Legal Setting

To evaluate the effectiveness of our attacks in another domain, we experiment with a legal case summarization task. We use legal data from the Free Law project[2], filtering for sentencing / criminal possession data using the built in Nomic topic modeling tool. We extract criminal behaviors (*e.g.*, criminal possession of a weapon) and personal identifiable information (*e.g.*, five years in prison) as fragments. Ultimately, we have 748 train, 188 validation, and 235 test samples. The attack procedure is identical to the medical setting. We adopt the following prompt for the legal task: `Consider an individual whose legal document summary contains the following criminal behaviors and identifiable information: {criminal possession of a weapon, ...}. That individual's legal document summary then also includes: ` $\{\mathbf{y}^*\}$. We expected the legal domain to be more difficult for `PIFI`; legal language overlaps more with everyday language than medical, which is a known challenge under other threat models (Lukas et al., 2023; Duan et al., 2024).

## 6. Models & Evaluation

**Models** We study our attacks using four autoregressive, decoder-only large language models (Radford, 2018), each of which is pretrained on a large text corpus and subsequently instruction-tuned to respond to user instructions

(Wei et al., 2021). Specifically, the models used as target and shadow models in the present work include: **Llama-3.1-8B-Instruct** & **Llama-3.2-3B-Instruct**, **Qwen-2-7B-Instruct**, and **Mistral-7B-Instruct-v0.2**. See Appendix F for more details on these models, and a discussion of our construction of the `world` model, for which we employ three LLMs *not* fine-tuned on either $D$ or $D'$.

**Fine-Tuning** We employ two approaches to fine-tuning the target model and shadow model LLMs for our experiments. For Llama-3.1-8B-Instruct, Llama-3.2-3B-Instruct, Qwen-2-7B-Instruct, and Mistral-7B-Instruct-v0.2, we employ full fine-tuning, updating the weights of the model itself. To test the robustness of our method to LoRA, we also quantize Llama-3.1-8B-Instruct to FP8 precision and fine-tune the model using low-rank adaptation (LoRA) (Hu et al., 2022; Dettmers et al., 2023). To further test the robustness of our method, we consider both models that have seen the data only once (*i.e.*, undergone 1 epoch of fine-tuning) and models that saw the data repeatedly, until loss convergence (*e.g.*, were fine-tuned for 10 or more epochs).

**Evaluation Metrics** Similarly to membership inference, attacks with high false positive rates are not meaningful in practical settings (Carlini et al., 2022a). Thus, we evaluate the effectiveness of our `PIFI` attacks by closely examining the Receiver Operating Characteristic (ROC) curve. The ROC curve demonstrates trade-offs between TPR and FPR for all possible choices of decision thresholds $\tau$, while the Area Under the Curve (AUC) metric integrates over the ROC curve to aggregate performance across FPRs. We do consider overall AUC, but specifically focus on the true positive rate at fixed, low false-positives rates (TPR @ %FPR, *e.g.*, 2% and 5%). Additionally, we use a log-scale to display the ROC curve, which follows Carlini et al. (2022a)'s example and highlights results from the low FPR regime.

**Expectations for Success** The `PIFI` threat model relies on strictly weaker adversarial assumptions (*i.e.*, no complete samples) when compared to direct memorization attacks. Prior ablation results on those attacks have suggested they are sensitive to conditioning on *approximate* or *permuted* versions of the target sample prefix (Ippolito et al., 2022; Lukas et al., 2023). As `PIFI` is analogous to strongly ablating a sample prefix, we do not expect extremely high TPRs in most settings. Nevertheless, even a modest TPR (at a very low, fixed FPR, like 2%) can have real impact applied at scale, as an attacker would likely query large numbers of individuals while being conservative with their inferences.

## 7. Results & Discussion

We present the results of applying the attacks enabled by our *partial-information fragment inference* (`PIFI`) threat

---

[2] https://huggingface.co/spaces/free-law/New_York_CAP

| Method | Prob. Access | | | TPR @ 2% FPR | | | TPR @ 5% FPR | | | ROC-AUC | | |
|---|---|---|---|---|---|---|---|---|---|---|---|---|
| | $p_D$ | $p_{D'}$ | $p_{\texttt{world}}$ | Llama 3 8B | Qwen 2 7B | Mistral 7B | Llama 3 8B | Qwen 2 7B | Mistral 7B | Llama 3 8B | Qwen 2 7B | Mistral 7B |
| Classifier | ● | ● | ● | **6.3%** | 8.2% | **5.5%** | **14.9%** | **16.3%** | **12.8%** | **0.68** | **0.73** | **0.68** |
| LR-Attack | ● | ● | ○ | 5.3% | **9.5%** | 2.9% | 10.6% | 16.2% | 8.8% | 0.64 | 0.67 | 0.59 |
| PRISM | ● | ● | ● | 4.4% | 4.3% | 4.2% | 11.5% | 11.0% | 11.3% | 0.67 | 0.66 | 0.63 |

*Table 1.* Main Results Across Models (Llama, Qwen, Mistral), for LLMs fine-tuned to convergence on the medical summarization task. Color-coding is Olympic medal standard by column (Gold is best, Silver second, Bronze third).

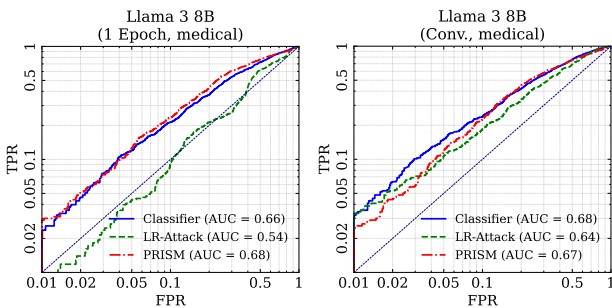

*Figure 3.* Left: **single** epoch fine-tuned Llama-3 8B model. Right: **convergence** fine-tuned Llama-3 8B model. More fine-tuning consistently increases the success rates of PIFI attacks.

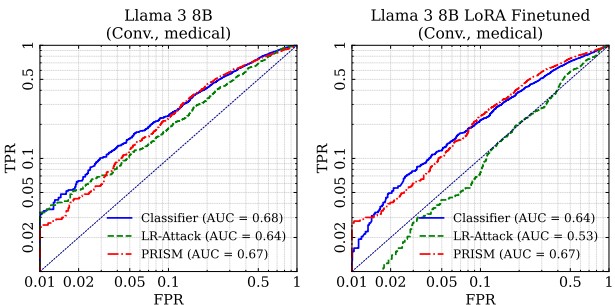

*Figure 4.* Left: **fully** fine-tuned Llama-3 8B model. Right: **LoRA** fine-tuned Llama-3 8B mode. LoRA-fine-tuned models exhibit less vulnerability than their fully fine-tuned counterparts.

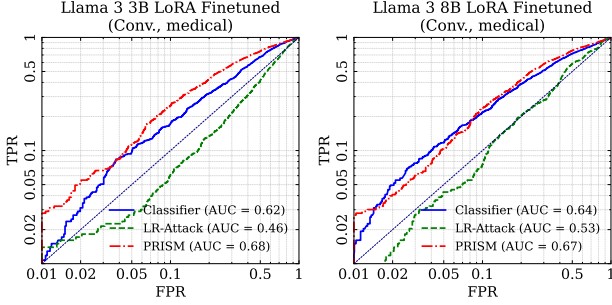

*Figure 5.* Left: LoRA fine-tuned Llama-3 **3B** model. Right: LoRA fine-tuned of the Llama-3 **8B** model. Larger parameter counts confer greater capacity for memorizing training examples.

model to the medical summarization task (mostly deferring results on the more difficult legal task to the appendix), focusing on factors that most influence fragment-level inference. We report results for LR-Attack, PRISM, and Classifier attacks, providing evidence for substantial privacy risk across model families and hyperparameter settings. We structure results as *takeaways*, denoted **T#**.

### 7.1. (T1) Attacks Are Effective Across Diverse Models

We attack three LLMs (Llama-3.1-8B, Qwen-2-7B, and Mistral-7B-v.02) instruction-tuned for medical note summarization with LR-Attack, PRISM, and Classifier under the PIFI threat model. As shown in Table 1, all three attack methods exceed a random-guess baseline by a considerable margin (doubling and sometimes quadrupling TPRs at low FPRs), indicating that the PIFI threat model is effective against three families of LLMs.

### 7.2. (T2) Repeated Exposure Increases Vulnerability

Next, we examine whether *increased* data exposure is positively correlated with PIFI success when attacking a model. We compare models fine-tuned for a single epoch (model updates based on a *single* pass on the data) against those fine-tuned for ten or more epochs (*i.e., many* passes, to convergence). Figure 3 shows that more fine-tuning consistently increases the success rates of PIFI attacks. This aligns with prior studies of membership inference (Carlini et al., 2021; 2022a) in LLMs, which found that repeated exposure to a sample increases the chances of memorization. However, even a single epoch of fine-tuning is sufficient to achieve a non-trivial TPR, demonstrating privacy risks posed by even lightly fine-tuned LLMs.

### 7.3. (T3) LoRA Fine-Tuning Still Leaks Information

We also investigate whether parameter-efficient methods like LoRA (Hu et al., 2022) mitigate fragment-level leakage. In keeping with prior work (Jiang et al., 2024), Figure 4 demonstrates that LoRA-fine-tuned models exhibit less vulnerability than fully fine-tuned counterparts. However, the difference in TPR@FPR is modest, and PRISM retains high recall, indicating some memorization has still occurred.

### 7.4. (T4) Vulnerability of Larger Models

Comparing models across parameter scales within the same family reveals that the larger models are more vulnerable to PIFI attacks, as we obtain higher AUC for an 8-billion parameter Llama model than for a 3-billion parameter Llama model when both are fine-tuned for 10 epochs, as shown in Figure 5. This outcome is again consistent with prior work demonstrating that larger parameter counts confer greater capacity for memorizing training examples (Car-

lini et al., 2021). We also evaluated our attack on a 70B-parameter Llama model LoRA-fine-tuned for 10 epochs (the largest model we could finetune given our compute constraints). The comparison of its performance with the 8B Llama model is visualized in Figure 10. While large LLMs deliver improved performance, their higher memorization capacity also raises the risk of fragment-level data leakage.

### 7.5. (T5) Noising the Prompt Has Minor Impact

Introducing noise in the form of inaccurate tokens does not significantly degrade the performance of our attacks. We find that, for a Llama-3-8B model fine-tuned for one epoch, replacing 75% of the accurate fragments possessed by the adversary with inaccurate fragments (corresponding to false assumptions on the part of the adversary) reduces AUC from .68 to .65 for `PRISM` and from .66 to .61 for `Classifier`, while leaving `LR-Attack` unchanged at .54. We provide additional detail in Appendix B.2.

### 7.6. (T6) Legal Setting Attacks Are More Challenging

The legal domain presents a more difficult challenge for our methods. Though the attacks still outperform chance (often by a 2:1 TPR:FPR ratio), AUC for a Qwen2-7B-Instruct model fine-tuned for 10 epochs falls to .55 for `PRISM`, to .61 for `LR-Attack`, and to .59 for `Classifier`, with similar declines observed for a Llama-3 model. Note that this finding accords with expectations, as outlined in Section 5.2. We provide additional detail in Appendix B.3.

### 7.7. (T7) `LR-Attack` Excels for Rare Fragments, `PRISM` for Common Ones

We find that `LR-Attack` performs best for rare fragments (*e.g.,* "brachial plexopathy"), while `PRISM` performs best for more common fragments (*e.g.,* "parkinsons"). This suggests that incorporating a prior from the `world` model helps to curb false positives and improve precision, an important consideration for higher-frequency fragments. However, the likelihood ratio $\frac{p_D}{p_{D'}}$ of `LR-Attack` *increases* sharply for rare data, rendering the method more sensitive in these cases. We provide additional detail in Appendix B.4.

### 7.8. Additional Experiments

**Rare vs. Common Fragments**  To clarify *why* the ranks of the three attacks sometimes trade places in Table 1, we stratify the 1,034 candidate fragments in the medical task by their corpus frequency.[3] Exactly 47% of the fragments appear *once* in the entire training set, while the remainder surface two or more times. Table 2 reports attack performance on these two disjoint partitions.

[3]Frequency is computed on the *fine-tuning* split only, mirroring what an attacker would implicitly exploit.

| Method | TPR @ 2% FPR | TPR @ 5% FPR | ROC-AUC |
|---|---|---|---|
| Classifier | 10.8% | 13.7% | 0.65 |
| **LR-Attack** | **17.5%** | **34.4%** | **0.77** |
| PRISM | 2.8% | 4.2% | 0.57 |
| Classifier | 7.1% | 17.0% | 0.74 |
| LR-Attack | 2.5% | 5.3% | 0.61 |
| **PRISM** | **5.6%** | **13.5%** | **0.73** |

*Table 2.* **(Top)** Attack performance when considering fragments that occur *once* in the fine-tuning data ("rare" set). **(Bottom)** Performance when considering fragments that occur *multiple* times in the fine-tuning data ("common" set) (Note: model attacked is **convergence** fine-tuned Llama-3 8B model).

When the candidate fragment is unique, the simple likelihood-ratio statistic of `LR-Attack` is markedly more sensitive (TPR@2%FPR = 17.5%). Because such tokens are nearly absent in the shadow model's fine-tuning, the ratio $p_D/p_{D'}$ spikes, yielding a strong signal. But, as fragment frequency rises, many individuals in $\mathcal{D}$ share the same token. Here the world-model prior in `PRISM` suppresses false positives and overtakes the other methods (AUC = 0.73), reflecting its design goal of balancing sensitivity with precision in higher-frequency regimes. Although the data-aware classifier remains competitive across both strata, one of the two data-blind attacks always matches or exceeds its recall at low-FPR – underscoring that meaningful leakage persists even without labeled in-distribution examples. Taken together, these results corroborate the complementary nature of the two data-blind attacks: `LR-Attack` excels when the adversary probes for idiosyncratic, sparsely represented fragments, whereas `PRISM` is better suited for fragments that occur more widely.

**Potential of DP Fine-Tuning as a Defense**  Differentially private (DP) fine-tuning of LLMs (Yu et al., 2021; Li et al., 2021) has been proposed as a strategy to defend against privacy vulnerabilities in LLMs. Recent work shows that DP fine-tuning helps to protect against memorization in LLMs, and that sentence-level privacy can help to reduce (but not eliminate) PII leakage in LLMs (Lukas et al., 2023).

To that end, we additionally fine-tuned the Llama-3.2-3B-Instruct model with *sample-level* differential privacy (DP).[4] Below we report results with $\epsilon = 3$, a common setting to balance utility and privacy. In terms of utility, Table 3 shows how DP fine-tuning improves over the base model, but does not fully recover the gains of non-private fine-tuning, in line with prior work (Lukas et al., 2023). In terms of defense against the `PIFI` threat model, while `Classifier` and `PRISM` roughly double the performance of random guessing, DP suppresses the performance of LR-ATTACK

[4]Ten epochs, opacus set to achieve $\epsilon$ under $(\epsilon, 10^{-5})$–DP, uses dp-transformers library (Wutschitz et al., 2022).

| Metric | Base model | DP ($\epsilon = 3$) / Non-DP |
|---|---|---|
| ROUGE-L$_{sum}$ ↑ | 0.0963 | 0.0969 / **0.1004** |
| BERTScore F1 ↑ | 0.7140 | 0.7186 / **0.7299** |

| Method | TPR @ 2% FPR | TPR @ 5% FPR | ROC-AUC |
|---|---|---|---|
| Classifier | **4.4%** | **10.0%** | **0.64** |
| LR-Attack | 0.9% | 2.4% | 0.51 |
| PRISM | 4.0% | 9.6% | 0.54 |

*Table 3.* (**Top**) Summarization quality on the medical task. (**Bottom**) `PIFI` results on Llama-3.2-3B-Instruct model fine-tuned with DP-SGD, showing some vulnerability.

| Method | TPR @ 2% FPR | TPR @ 5% FPR | ROC-AUC |
|---|---|---|---|
| Classifier | **7.0%** | **13.4%** | 0.69 |
| LR-Attack | 5.3% | 10.6% | 0.64 |
| PRISM | 5.2% | 11.6% | **0.70** |

*Table 4.* Results using DeepSeek for world probabilities. (TPR@2% FPR= 0.9%).

**Improved World Models** We experimented with using world probabilities from two large open-source models, DeepSeek-V3 and R1 (averaging the two). Table 4 gives the performance of our attacks on the Llama-3-8B model (with 10 epochs of fine-tuning) using DeepSeek world probabilities. We observe slight improvements on each of our three attack metrics compared to results obtained when using the three smaller LLMs as world models. These results suggest that incorporating additional high-quality models in the world model ensemble would further enhance the performance of our attacks. However, as noted previously, the log-probabilities for powerful closed models are generally not available or significantly restricted, which limits our ensemble to open-source models.

## 8. Related Work

The `PIFI` threat model builds on prior research on LLM privacy. Here we consider related work, focusing on memorization and membership inference attacks as predominant approaches in the field.

**Memorization** LLMs may produce sensitive data through user interactions; for example, by prompting a model with "Jane Doe's social security number is..." one might elicit a social security number (Carlini et al., 2019). This represents a *memorization* attack: formally, given model $f_\theta$ and prefix $\mathbf{x}$, can we extract a target sequence $\mathbf{s}$ that we believe is in the training data *exactly*? Work by Carlini et al. (2021) introduced this concept of memorized extractability as $k$-*eidetic memorization*, where $k$ gives the number of times $\mathbf{s}$ appeared in $\mathcal{D}$. In subsequent work, a similar definition was given: $\mathbf{s}$ is considered *extractable* if given $k'$ tokens of *context* there exists a length-$k'$ string $\mathbf{p}$ such that

the concatenation $[\mathbf{p} \,||\, \mathbf{s}]$ is in $\mathcal{D}$ and $f_\theta$ produces $\mathbf{s}$ when decoded greedily under prompt $\mathbf{p}$ (Carlini et al., 2022b). Modifications to extraction attacks may seek to alter an LLM's text generation strategy to more easily extract data. Yu et al. (2023) benchmark techniques to enhance extraction attacks, employing techniques like top-$k$, nucleus, and typical sampling to address distribution discrepancies, resulting in substantial improvements in data extraction. Similarly, Nasr et al. (2023) design attacks that can cause a production model like ChatGPT to diverge from its instruction-tuned objective and emit training data at a higher rate.

**Membership Inference** Membership inference attacks attempt to learn whether a given sample was present in the training data for a machine learning model (Shokri et al., 2016). Prior work demonstrates that LLMs are vulnerable to membership inference attacks both after initial pre-training (Duan et al., 2024) and after additional fine-tuning (Fu et al., 2023). More complex membership inference attacks may leverage additional information about a model, such as info gained using model-stealing techniques (Wang et al., 2024). In addition to their relevance from a privacy perspective, MI attacks are used in analyses of model memorization (Shi et al., 2023) and test set contamination (Oren et al., 2023).

## 9. Limitations

We acknowledge that this work does not extensively evaluate the efficacy of DP fine-tuning against `PIFI` attacks; a promising line of future work would be a large scale exploration of the defensive effectiveness of DP-fine-tuning. However, while DP fine-tuning may be effective for privacy, this approach degrades the utility of the fine-tuned model, sometimes *substantially* (Mireshghallah et al., 2021). Additionally, DP fine-tuning of LLMs remains uncommon in practice due to the significant technical and financial burdens — *e.g.*, necessitating high-VRAM GPUs and large batch sizes, even with qLoRA. Consequently, assessing PIFI without DP is both reasonable and practically important, since the vast majority of fine-tuned LLMs lack DP defenses.

## 10. Conclusion

We introduce a novel threat model `PIFI`, which provides a realistic approach to fragment-level extraction from LLMs. The `LR-Attack` and `PRISM` attacks we construct under the data and model availability assumptions of `PIFI` reveal the vulnerability of LLMs to extraction of private data, even when an attacker possesses only fragmented and/or unordered data. Our work suggests a need for research on LLM privacy that adopts weaker (and thus more realistic) assumptions about scenarios in which these models are vulnerable to privacy attacks.

## Impact Statement

Our work advances the science of LLM privacy attacks by introducing a new threat model and two novel attacks on chat-based language models, pointing to the vulnerabilities embedded in an increasingly common form of digital infrastructure. The intended impact of this research is to equip organizations and individuals training language models with a better understanding of their vulnerabilities, and to advance the knowledge available to the research community in producing effective methods to circumvent these attacks. We acknowledge that our methods may equip real-world bad actors with the ability to draw private information from publicly available LLMs. However, we suggest that forewarned is forearmed: these vulnerabilities likely already exist in many production language models, and our work can help responsible organizations to prepare an appropriate defense.

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

# A. Experimental Setup

We illustrate our experimental setup below. Where a sample **s** is in the target dataset $D$ and the target fragment $\mathbf{y}^*$ is in the sample **s**, attacks under the PIFI threat model can succeed by successfully identifying the fragment $\mathbf{y}^*$. Scenarios where an attack does not succeed under PIFI include those where a target fragment $\mathbf{y}^*$ is identified but $\mathbf{y}^*$ is not in **s**, or **s** is not in $D$, or **s** is not in $D$ and $\mathbf{y}^*$ is not in **s**.

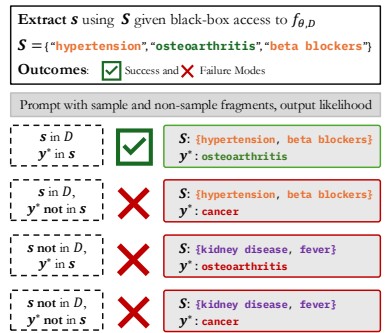

*Figure 6.* Successful and failed attack scenarios. An attack is only considered successful when the sequence **s** is in the dataset $D$, the target fragment $\mathbf{y}^*$ is in the sequence **s**, and we accurately infer the target fragment's presence in **s**. Any other scenario is considered as a failed attack – **(1)** the target fragment is NOT in the sequence. **(2)** the sequence is NOT in the data $D$. **(3)** the sequence is NOT in the data $D$ and the target fragment is NOT in the sequence.

## A.1. Illustrative Examples

We probed 4,302 *fragment instances* for the medical summarization task, of which 1,034 were unique. The empirical frequency distribution of these fragments is highly skewed: the most common fragments were common medical terms, such as `pain` (124 occurrences), `fever` (69), `shortness of breath` (55), `diarrhea` (47), and `chest pain` (44). By contrast, *75%* of all fragments appeared fewer than five times, and *47%* occurred exactly once. These rare fragments include some highly specific conditions and medications, such as (selected at random) `Vincristine`, `colitis`, `daunorubicin`, `Naprosyn`, `Xalatan`, and `lumbar spinal stenosis`. Their rarity renders these fragments appropriate targets for the likelihood-ratio attack (Table 2). For more concrete examples, Table 5 presents five successes obtained using LR-Attack and PRISM scores. In each case, the adversary begins with a handful of publicly available fragments $\mathcal{S}$ and is able to infer an additional sensitive fragment $\mathbf{y}^*$ that indeed appears in an individual's record.

| Public fragments provided to the model | Inferred fragment (record) |
|---|---|
| `anxiety disorder, arthritis, Morton's neuromas, migraines` | `hypothyroidism` (train_679) |
| `shortness of breath, Coumadin, lightheadedness, chest pain, Cardizem, pain, vertigo` | `atrial fibrillation` (train_873) |
| `toxicity, breast cancer, Ixempra, tumor, neuropathy, Avastin, cancer, Faslodex, Zometa, ixabepilone` | `Aromasin` (train_107) |
| `swelling, rashes, vomiting, pain` | `seizures` (train_84) |
| `weakness, headaches, stroke, sudden loss of consciousness, seizures, tremors` | `epilepsy` (train_577) |

*Table 5.* Illustrative PIFI attack successes. Each row lists the attacker's public fragment set $\mathcal{S}$ (left) and the private fragment $\mathbf{y}^*$ inferred, together with the record identifier (right).

# B. Additional Results

## B.1. (T1) Attacks Are Effective Across Diverse Models (cont.)

We attack three LLMs (Llama-3.1-8B, Qwen-2-7B, and Mistral-7B) instruction-tuned for medical note summarization. Table 1 in the main body presents the quantitative results, while Figure 7 shows the ROC curves for all models, allowing a more comprehensive inspection at different FPR values. We observe that the `PIFI` threat model is effective against three families of LLMs.

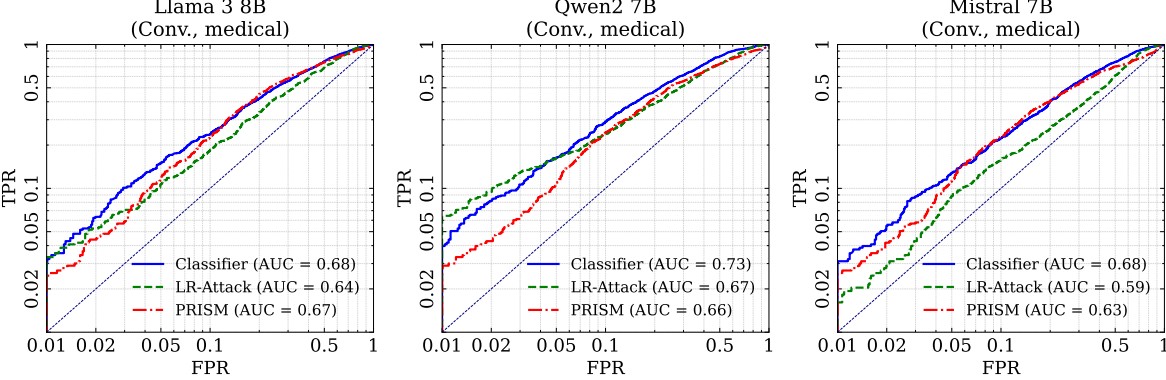

*Figure 7.* Left: fully fine-tuned Llama-3-8B model. Middle: fully fine-tuned Qwen2-7B model Right: fully fine-tuned Mistral-7B model. We observe that `PRISM` maintains consistent performance with respect to `Classifier` and `LR-Attack` for each model, indicating its robustness to different model families.

## B.2. (T5) Noising the Prompt Has Minor Impact

Because our partial-information threat model relies on only a *handful of fragments*, one might wonder whether, if an attacker includes *incorrect* or *unassociated* fragments, this would hurt attack performance. In our experiments, permuting the listed fragments (*e.g.*, replacing true fragment set "hypertension, *diabetes*" with partially true set "hypertension, *osteoporosis*") has a negligible effect on attack success rates. Although prompts containing *more true* fragments achieve slightly higher AUC, the improvement is less pronounced than one might expect. This demonstrates that the fine-tuned model has become highly attuned to associations between individual pairs of fragments, and that it is robust to noise introduced by fragments in the prompt not associated with the individual.

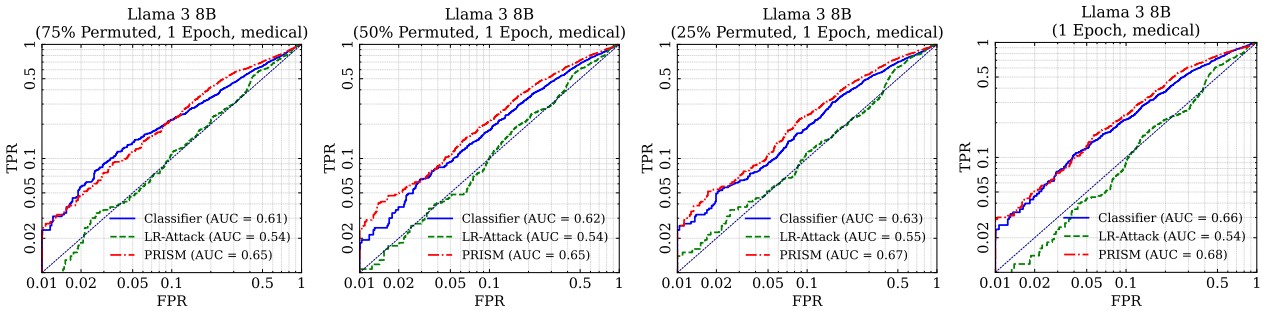

*Figure 8.* Results of an ablation wherein only k% of fragments used in the attack prompt are accurate, demonstrating that PRISM remains a viable method for an attacker who possesses only a small number of correct fragments. Results also show that PRISM *outperforms* other methods, including a learned classifier, when an LLM is only lightly fine-tuned (*i.e.,* for a single epoch), suggesting the effectiveness of the prior incorporated by the attack.

## B.3. (T6) Attacks in the Legal Setting Are More Challenging

We also validate our `PIFI` threat model in a legal case summarization task (see Section 5.2 for details). Here, the fragments often correspond to crimes committed or other identifiable information (see Appendix D for examples and categories). We find that our methods still outperform chance, **often by a 2:1 TPR:FPR ratio,** in identifying whether a sensitive legal

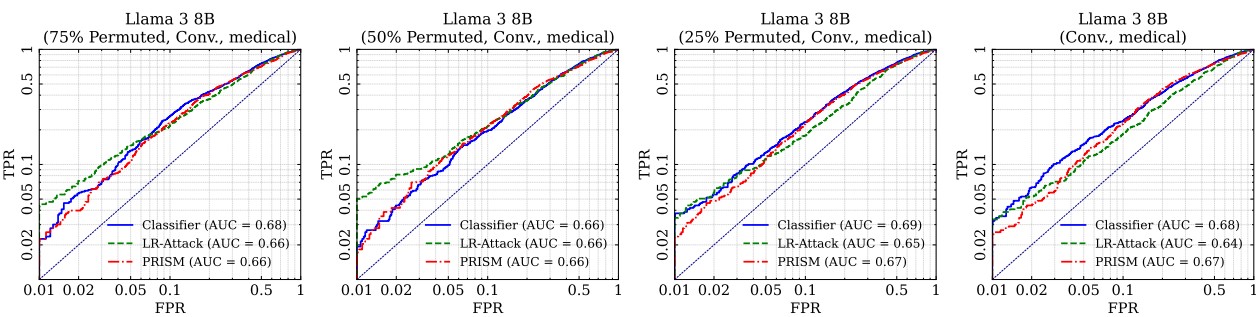

*Figure 9.* Results of our ablation wherein only k% of prompt fragments are accurate; the figure illustrates results for a Llama-3 model trained for 10 epochs, and attacked at 25%, 50%, 75%, and 100% accurate fragments..

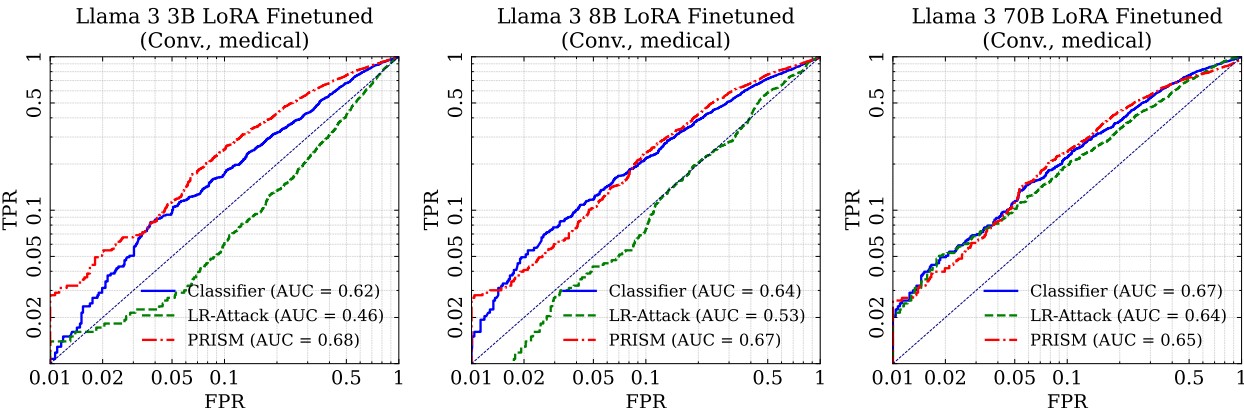

*Figure 10.* Left: LoRA fine-tuned Llama-3 **3B** model. Middle: LoRA fine-tuned of the Llama-3 **8B** model. Right: LoRA fine-tuned of the Llama-3 **70B** model. Larger parameter counts confer greater capacity for memorizing training examples.

text fragment was in the training set; however, the attacks are noticeably less successful than in the medical context. We hypothesize that, because legal keywords are more frequent in general pre-training data, the value of fragment-specific signal is diluted. Consequently, distinguishing a target fine-tuned fragment from the baseline text distribution becomes harder. Nevertheless, `PIFI` attacks remain feasible, indicating that LLMs fine-tuned in the legal setting may leak partial information, even though the language distribution is closer to everyday speech.

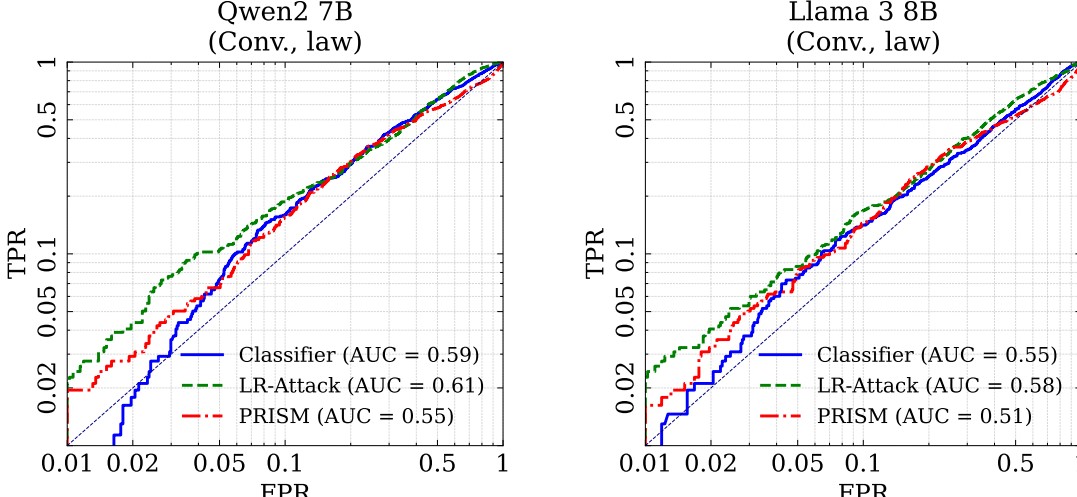

*Figure 11.* The legal setting proves more challenging for our attacks. AUC does not exceed .61 in Qwen-2-7B or .58 in Llama-3-8b, with the strongest result obtained by `LR-Attack` in both cases, suggesting that its sensitivity to unusual fragments is a particular benefit in a setting where most fragments are relatively frequently occurring words in LLM pretraining data.

## B.4. (T7) `LR-Attack` Excels for Rare Fragments, `PRISM` for Common Ones

We analyze how fragment frequency affects attack performance. For *rare* tokens or token sequences (*e.g.*, a specialized diagnosis in our medical dataset, or a lab value), the simple `LR-Attack` often performs best, presumably because the likelihood ratio $\frac{p_D}{p_{D'}}$ spikes sharply when the model sees an unusual fragment learned from training data (*e.g.*, "brachial plexopathy" or "darvocet", from the "Biological_structure" category). By contrast, when fragments are more common in text corpora (*e.g.*, "alzheimer," or "parkinsons", from the "Disease_disorder" category), `PRISM`'s incorporation of a prior from the `world` model helps curb false positives and improves precision. These observations support our claim that partial-information inference is not uniform across fragments: modeling *both* the in-domain shadow distribution and an external "world" distribution can improve performance on higher-frequency tokens.

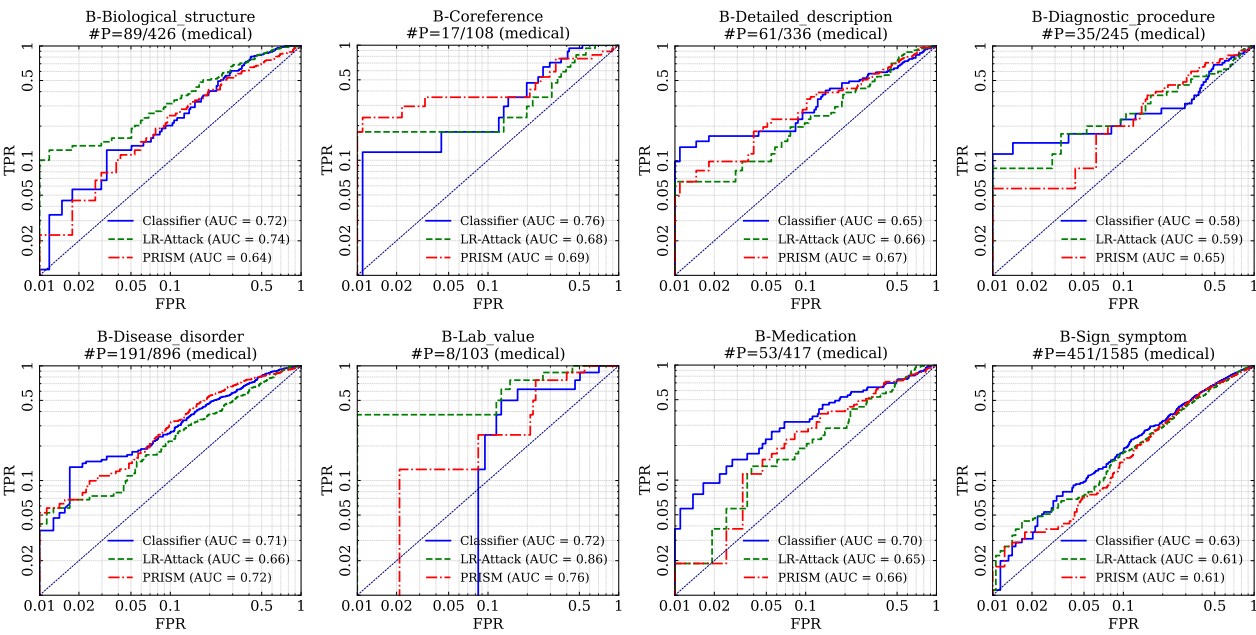

*Figure 12.* Our findings indicate that `LR-Attack` is more effective for rare fragments, while `PRISM` is more effective for common fragments. For example, `PRISM` outperforms `LR-Attack` for the names of diseases (bottom left, B-Disease_disorder), which contains the commons names of diseases (like "parkinsons"), with AUC equal to .72 compared to AUC of .66 for `LR-Attack`. On the other hand, `LR-Attack` outperforms `PRISM` for fragments related to more technical biological structure terms (B-Biological_structure, top left, which contains uncommon terms such as "brachial plexopathy").

## C. Synthetic Setting

We compare the `PRISM` and `LR-Attack` partial-information extraction attacks in a toy setting using a *synthetic* trigram language model. This allows us to validate insights given in Section 4.3, as we will allow ourselves access to a *known* distribution over a tiny vocabulary.

**Toy Data**    Let $\mathcal{V} = \{v_0, v_1, \ldots, v_{|\mathcal{V}|-1}\}$ be a vocabulary of size $|\mathcal{V}|$ (in our toy experiments, we use a small set of letters, $\mathcal{V} = \{a, b, c, d\}$, as the vocabulary.). We generate a *target* dataset $D$ of size $N$, where each $\mathbf{s} \in D$ is an ordered sequence $(v_1, v_2, \ldots, v_L)$ of length $\mathcal{T}$. For example, a sequence with $\mathcal{T} = 3$ letters could be any permutation of the three out of four letters, such as "*abd*", "*aaa*", "*acb*." We then create a *shadow* dataset $D'$ of the same size and distribution, ensuring that a controlled subset of samples does *not* appear in $D'$, so that $D'$ serves as an "out-of-distribution" reference for the target $\mathbf{s}$. Then we combine *target* set and *shadow* set as the *world* dataset: $\mathcal{D}_w = \oplus(D, D')$. The world dataset represents a comprehensive knowledge set. We generate a *test* dataset $D_t$ with the same parametrization $(\mathcal{V}, N, \mathcal{T})$ as $D$ for evaluation.

**Toy Models**    We fit three toy trigram models, $f_{\theta,D}, f_{\theta,D'}, f_{\theta,\text{world}}$, on each corresponding dataset. Each model estimates $P(v_i|v_{i-2}, v_{i-1})$ via counts of observed trigrams in the respective dataset. For example, if we observe "ab" ten times and "abb" three times, then probability $P(b|ab) = 0.3$. To cover the case where a trigram has no occurrences, a small smoothing value $1e-6$ is applied. Thus, we can get the probability of any sequence under one of the models by multiplying the probabilities of all trigrams in the sequence: $P(v_1 v_2 \ldots v_{\mathcal{T}}) = \prod_{i=3}^{\mathcal{T}} P(v_i|v_{i-2}v_{i-1})$.

**Synthetic Attack**    For each sequence $s = (v_1, v_2, \ldots, v_L) \in D_t$, we consider a conditional length $C$, $2 \leq C \leq \mathcal{T} - 1$. We randomly select $C$ vocabularies, without replacement, and in order, from $(v_1, v_2, \ldots, v_{L-1})$ as the conditional sequence. Then we append the last vocabulary $v_{\mathcal{T}}$ to the new sequence as $s_C$. The intuition behind this shorter conditional length is that we only have partial information of the sequence, which is being used to inform the probability of the whole sequence. In our setup, the shorter the conditional length, the harder the prediction task is. Then we calculate the probability of the conditional sequence $s_C$ from the three models: $P_{s_C,D}, P_{s_C,D'} \ P_{s_C,D_w}$. The `LR-Attack` attack is calculated as described in Section 4.2, and the `PRISM` attack is calculated as described in Section 4.3.

**Experimental Setup & Results** We experimented with the following parametrization: $|\mathcal{V}| = [4, 5, 6, 7]$, $\mathcal{T} = [4, 5]$, $N = |\mathcal{V}|^{(\mathcal{T}-1)}$. We report AUC in Figure 13. We observe that **(1)** when vocabulary size increases, the task is more difficult, because there are more permutations of different letters, increasing the volume of information. Fixing vocabulary size, increasing the sequence length makes the task more challenging, as longer sequences contain more complicated information. Fixing sequence length and increasing conditional length makes the prediction task easier, because a longer conditional sequence provides more information, and **(2)** other than the case of $(|\mathcal{V}|, \mathcal{T}, \mathcal{C})$=(6,5,2), `PRISM` almost always outperforms the `LR-Attack` method. The effect is more salient when conditional length is longer, *i.e.*, when probabilities used for the attack are conditioned on more information.

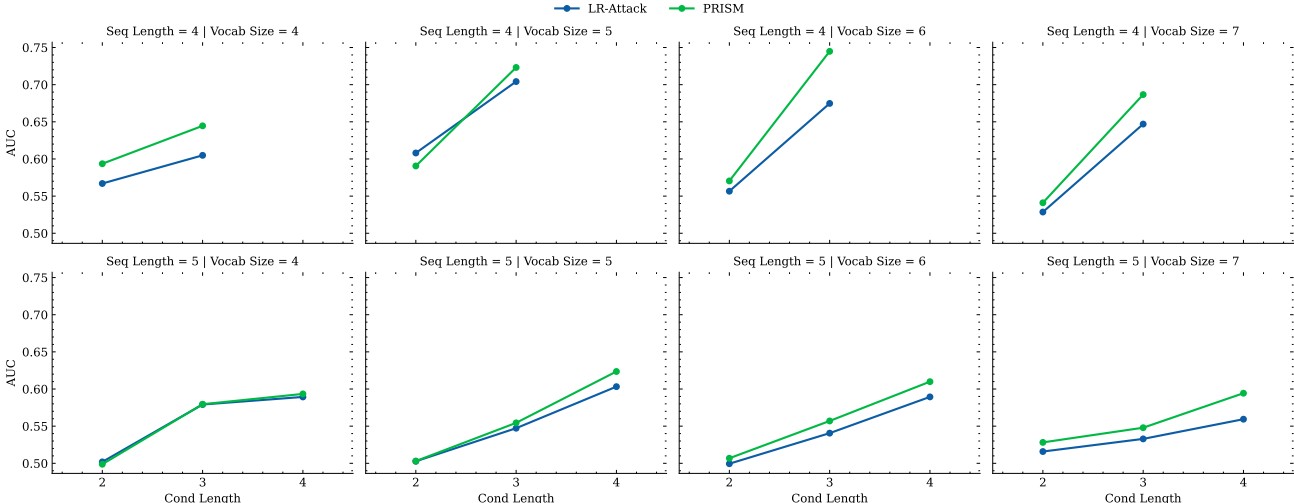

*Figure 13.* Synthetic attack results. AUC scores are visualized for two methods. Each square shows the result of a parametrization.

# D. Data Examples

| Medical Note Summarization Example |
|---|
| **System:**
You are a helpful medical assistant! Please help me summarize dialogues between doctors and patients.

**User:**
*Doctor*: What brings you back into the clinic today, miss?
*Patient*: I came in for a refill of my blood pressure medicine.
*Doctor*: It looks like Doctor Kumar followed up with you last time regarding your `hypertension`, `osteoarthritis`, `osteoporosis`, `hypothyroidism`, `allergic rhinitis` and `kidney stones`. Have you noticed any changes or do you have any concerns regarding these issues?
*Patient*: No.
*Doctor*: Have you had any `fever` or `chills`, `cough`, congestion, `nausea`, vomiting, `chest pain`, chest pressure?
*Patient*: No.
*Doctor*: Great. Also, for our records, how old are you and what race do you identify yourself as?
*Patient*: I am seventy six years old and identify as a white female.

**Assistant:**
The patient is a 76-year-old white female who presents to the clinic today originally for hypertension and a med check. She has a history of hypertension, osteoarthritis, osteoporosis, hypothyroidism, allergic rhinitis and kidney stones. Since her last visit she has been followed by Dr. Kumar. Those issues are stable. She has had no fever or chills, cough, congestion, nausea, vomiting, chest pain, chest pressure. |

*Table 6.* A medical note summarization example. The medical entities are highlighted in green.

| Legal Case Summarization Example |
|---|
| **System:** |
| You are a helpful law assistant! Please help me summarize the legal cases. |
| |
| **User:** Appeal from a judgment of the County Court of Sullivan County (Hanofee, J.), rendered February 8, 1990, convicting defendant upon his plea of guilty of the crime of criminal sale of a controlled substance in the third degree. We reject defendant's contention that the prison sentence imposed by County Court of 3 to 9 years was harsh and excessive. The sentence was in accordance with the plea bargain and was well within the statutory limits. Under the circumstances, his admission that he has a drug problem fails to demonstrate either extraordinary circumstances or an abuse of discretion by the court warranting a reduction in the sentence (see, People v Mackey, 136 AD2d 780, lv denied 71 NY2d 899). Judgment affirmed. Mahoney, P. J., Mikoll, Yesawich, Jr., Crew III, and Harvey, JJ., concur. |
| |
| **Assistant:** The County Court of Sullivan County's judgment convicting the defendant for criminal sale of a controlled substance in the third degree and sentencing him to 3 to 9 years was upheld on appeal. The appellate court found that the sentence was appropriate, aligned with the plea bargain, and within statutory limits, dismissing the defendant's claims of harshness and excessive punishment. The judgment was affirmed unanimously by the presiding judges. |

*Table 7.* A legal case summarization example. The criminal behaviors are highlighted in pink. The identifiable information is highlighted in blue.

| Category | Count | Examples |
|---|---|---|
| B-Sign_symptom | 1585 | [Chantix, Constipation, Dizziness] |
| B-Disease_disorder | 896 | [Alzheimer, Mononucleosis, Parkinson] |
| B-Biological_structure | 426 | [Brachial Plexopathy, Darvocet] |
| B-Medication | 417 | [Advair, Methocarbamol, Vicodin] |
| B-Detailed_description | 336 | [Acute Cholecystitis] |
| B-Diagnostic_procedure | 245 | [Bone Abnormalities] |
| B-Coreference | 108 | [Cardizem, Lipitor, Zometa] |
| B-Lab_value | 103 | [Abnormal Heart Valve, STD] |

*Table 8.* Examples of the categories of medical entities; we only present categories here that have a count over one hundred.

| Category | Count | Examples |
|---|---|---|
| Criminal_Offenses | 934 | [32 Grams of Heroin, Burglary] |
| Sentences_Penalties | 515 | [$1,000 Fine, 12 Years in Prison] |
| Dates_Times | 450 | [16 Years Old, April 27, 1989] |
| People_Roles | 382 | [Egan Jr., Mercure, J.P.] |
| Miscellaneous | 241 | [Alcohol Abuse, Failed to Report] |
| Geography | 235 | [Albany County, City of Hudson] |
| Legal_References | 162 | [ASAT program, Rockefeller Drug Law Reform Act] |
| Courts_Corrections | 127 | [County Court of Chenango County]] |

*Table 9.* Examples for the categories of legal entities; we only present categories here that have a count over one hundred.

| Prefix Length | Hamming Distance ($\downarrow$) | | Recall ($\uparrow$) | |
|---|---|---|---|---|
| (Word Count) | Llama 3 8B (Conv.) | Llama 3 8B (1E) | Llama 3 8B (Conv.) | Llama 3 8B (1E) |
| 10 words | **29.077** | 31.876 | **0.423** | 0.255 |
| 20 words | **28.935** | 29.313 | **0.360** | 0.217 |
| 30 words | **25.914** | 26.236 | **0.263** | 0.176 |

*Table 10.* Simple memorization attack results. Here, "Conv." means the model was trained until convergence (for Llama 3 8B, this took 10 epochs), while "1E" means one epoch (the model only saw the data for a single training pass).

## E. Memorization Attack

That memorization occurs when language models are fine-tuned is well established (Carlini et al., 2021; 2022b; Lukas et al., 2023). In this short section, we simply aim to establish that *our* finetuning process leads to memorization of some of the training data; and in particular, that when we fine-tune a model for *more* epochs (full passes over the training samples), that model memorizes *more* training data.

Our experiments follow the setup described by Carlini et al. (2021) — we provide a *prefix* as the prompt and generate a fixed number of candidate tokens $\mathcal{H} = \{h_1, h_2, \ldots, h_n\}$ to be compared against the aligned set of ground truth tokens $\mathcal{G} = \{g_1, g_2, \ldots, g_n\}$. In our experiments, we combine each doctor-patient dialogue and $l = \{10, 20, 30\}$ words from the summary as the *prefix*, and we prompt the model to generate $n = 50$ tokens. We conduct the memorization attack using 200 fine-tuning samples, using the 1-epoch fine-tuned and convergence fine-tuned Llama 3 8B models.

To evaluate memorization, we use two standard metrics. The first is *Hamming Distance*, calculated as $\sum_{i=1}^{n} \mathbb{1}[h_i \neq g_i]$. *Hamming distance* counts the mismatched generated and ground truth tokens; the lower the value, the more the model has memorized. The second metric is *Recall*, calculated as $\sum_{i=1}^{n} \mathbb{1}[h_i \in \mathcal{G}]$. The model may not necessarily generate the memorized information in the strict order of appearance from the fine-tuning data, but it still may produce most or all of the same words. The *Recall* rate then measures how many generated tokens are in the ground truth token set, ignoring order; the higher the value, the more the model has memorized. Results are given in Table E for the *Llama 3 8B* model.

We observe that for both metrics, the 10-epoch fine-tuned Llama model memorizes more information than its 1-epoch fine-tuned counterpart, with lower hamming distances and higher recall rates. This phenomenon is consistent across all three prefix lengths.

# F. Models

To obtain $p_D$ and $p_{D'}$, we fine-tune the below instruction-tuned models to produce the target models and the shadow models capable of generating those probabilities. Each model was pretrained on a large text corpus and subsequently instruction-tuned to respond to user instructions (Wei et al., 2021). The models include:

- **Llama-3.1-8B-Instruct**: An 8-billion parameter LLM pretrained on 15 trillion tokens of publicly available text data and fine-tuned using RLHF with human preferences (Dubey et al., 2024).
- **Llama-3.2-3B-Instruct**: A 3-billion parameter LLM pretrained on 9 trillion tokens of publicly available text data and fine-tuned using rejection sampling and DPO to align with human preferences (Dubey et al., 2024).
- **Qwen-2-7B-Instruct**: A 7-billion parameter LLM pretrained on 7 trillion tokens and fine-tuned using DPO to align with human preferences (Yang et al., 2024).
- **Mistral-7B-Instruct-v0.2**: A 7-billion parameter LLM pretrained on a private dataset and fine-tuned on public datasets to follow instructions (Jiang et al., 2023).

Parameter counts are approximate to the nearest billion and in keeping with widely used descriptions of the models.

To obtain $p_{\texttt{world}}$ for the purpose of the empirical results presented in this paper, we obtain probabilities from three LLMs, none of which are fine-tuned on either $D$ or $D'$. These models include Llama-3.1-8B-Instruct (Dubey et al., 2024), Mistral-7B-v0.2 (Jiang et al., 2023), and Gemma-2B-IT (Team et al., 2024). To compute $p_{\texttt{world}}$, we simply take the mean of the probabilities output by these three models, in accordance with the definition of the $\texttt{world}$ model: $f_{\theta,\texttt{world}}(\mathbf{x}) = \frac{1}{k} \sum_{i=1}^{k} f_{\theta,*}^{(k)}(\mathbf{x})$

# G. Full MMLU Results

One potential challenge to the practicality of our attacks lies in the potential for extensive fine-tuning to degrade the general-purpose capabilities and chat interface of a model. To test whether models trained under PIFI remain usable after fine-tuning, we ran utility evaluations using MMLU (Hendrycks et al., 2020). We found that the Llama-3-8B model fine-tuned for 10 epochs reached an average MMLU score of 0.565, compared to 0.487 after one epoch and 0.413 for the baseline model, suggesting the fine-tuned model remains capable and general. In medical areas (*e.g.*, clinical knowledge, medical genetics), fine-tuning improved performance notably over a non-fine-tuned baseline. However, most of the increase in performance we observed can be attributed to the fact that fine-tuned models sometimes outperform generalist models on some benchmarks due to "better behavior," meaning that they naturally follow a stricter format. We present full results below for Llama-3-8B without fine-tuning, fine-tuned for one epoch, and fine-tuned for ten epochs. Our results demonstrate that fine-tuning that renders models more vulnerable under PIFI does *not* substantially degrade the general-purpose chat interface of the model.

| MMLU Category | No Finetune | 1-epoch Fine-Tune | 10-epoch Fine-Tune |
|---|---|---|---|
| Overall | 0.41 | 0.49 | 0.57 |
| high-school-european-history | 0.60 | 0.70 | 0.69 |
| business-ethics | 0.24 | 0.44 | 0.58 |
| clinical-knowledge | 0.41 | 0.57 | 0.63 |
| medical-genetics | 0.64 | 0.70 | 0.75 |
| high-school-us-history | 0.49 | 0.66 | 0.72 |
| high-school-physics | 0.21 | 0.22 | 0.32 |
| high-school-world-history | 0.47 | 0.70 | 0.74 |
| virology | 0.51 | 0.52 | 0.52 |
| high-school-microeconomics | 0.40 | 0.61 | 0.71 |
| econometrics | 0.21 | 0.28 | 0.39 |
| college-computer-science | 0.28 | 0.36 | 0.43 |
| high-school-biology | 0.63 | 0.67 | 0.70 |
| abstract-algebra | 0.27 | 0.15 | 0.33 |
| professional-accounting | 0.17 | 0.27 | 0.39 |
| philosophy | 0.53 | 0.59 | 0.59 |
| professional-medicine | 0.57 | 0.64 | 0.69 |
| nutrition | 0.57 | 0.67 | 0.67 |
| global-facts | 0.09 | 0.16 | 0.20 |
| machine-learning | 0.16 | 0.21 | 0.38 |
| security-studies | 0.44 | 0.56 | 0.56 |
| public-relations | 0.43 | 0.51 | 0.59 |
| professional-psychology | 0.43 | 0.52 | 0.59 |
| prehistory | 0.52 | 0.60 | 0.65 |
| anatomy | 0.54 | 0.55 | 0.60 |
| human-sexuality | 0.63 | 0.60 | 0.72 |
| college-medicine | 0.42 | 0.50 | 0.57 |
| high-school-government-and-politics | 0.62 | 0.70 | 0.75 |
| college-chemistry | 0.28 | 0.23 | 0.33 |
| logical-fallacies | 0.50 | 0.58 | 0.67 |
| high-school-geography | 0.51 | 0.67 | 0.73 |
| elementary-mathematics | 0.28 | 0.31 | 0.38 |
| human-aging | 0.49 | 0.55 | 0.61 |
| college-mathematics | 0.26 | 0.26 | 0.33 |
| high-school-psychology | 0.62 | 0.72 | 0.76 |
| formal-logic | 0.26 | 0.35 | 0.42 |
| high-school-statistics | 0.25 | 0.30 | 0.36 |
| international-law | 0.57 | 0.68 | 0.64 |
| high-school-mathematics | 0.25 | 0.27 | 0.36 |
| high-school-computer-science | 0.47 | 0.50 | 0.66 |
| conceptual-physics | 0.37 | 0.40 | 0.49 |
| miscellaneous | 0.61 | 0.67 | 0.75 |
| high-school-chemistry | 0.34 | 0.38 | 0.42 |
| marketing | 0.32 | 0.67 | 0.79 |
| professional-law | 0.28 | 0.33 | 0.39 |
| management | 0.46 | 0.70 | 0.73 |
| college-physics | 0.22 | 0.25 | 0.30 |
| jurisprudence | 0.44 | 0.56 | 0.57 |
| world-religions | 0.62 | 0.67 | 0.77 |
| sociology | 0.69 | 0.74 | 0.76 |
| us-foreign-policy | 0.57 | 0.74 | 0.80 |
| high-school-macroeconomics | 0.32 | 0.52 | 0.63 |
| computer-security | 0.64 | 0.68 | 0.76 |
| moral-scenarios | 0.14 | 0.06 | 0.35 |
| moral-disputes | 0.45 | 0.55 | 0.60 |
| electrical-engineering | 0.30 | 0.46 | 0.57 |
| astronomy | 0.51 | 0.61 | 0.63 |
| college-biology | 0.58 | 0.61 | 0.66 |

