# OpenReview forum: "Fragments to Facts: Partial-Information Fragment Inference from LLMs"
_ICML.cc/2025/Conference — ICML 2025 poster_

### Official Review · Reviewer_rGXA · 2025-03-12

**Overall Recommendation:** 4

**Summary:**

The paper proposes a new privacy threat for fine-tuned LLMs called “Partial-Information Fragment Inference” (PIFI). The authors show that even if an attacker only knows a few scattered keywords about someone’s data (e.g., certain medical terms), they can still prompt the model to uncover additional, sensitive details. They develop two simple yet effective attacks (a likelihood-ratio approach and one called “PRISM”). Experiments in both medical and legal settings confirm that fine-tuned models can leak private fragment-level information, underscoring the need for more robust privacy defenses.

**Claims And Evidence:**

Yes

**Essential References Not Discussed:**

As far as my knowledge no.

**Experimental Designs Or Analyses:**

Experimental design is sound.

**Methods And Evaluation Criteria:**

Yes

**Other Comments Or Suggestions:**

Please look at above.

**Other Strengths And Weaknesses:**

Strength:

* A realistic assumptions on adversary attacks with partial fragments.
* Acknowledging different ablations like more number of epochs, comparing models with different parameter sizes, etc.

Weakness:
 * Classifier is considered as the baseline for experiments. Are there any other method in the literature that you can make the comparison with?
* You claim the TPR at low FPR isn’t random and is truly meaningful, matching a classifier’s performance. But how do we know the model isn’t just capturing domain-wide patterns (like “cold” usually co-occurring with “cough”) rather than genuine memorization at the individual level? Is there an experiment showing it’s not merely picking up generic associations from the entire dataset? That part still confuses me when considering a single datapoint’s data.

**Questions For Authors:**

Question1: Classifier is considered as the baseline for experiments. Are there any other method in the literature that you can make the comparison with?
Question 2: You claim the TPR at low FPR isn’t random and is truly meaningful, matching a classifier’s performance. But how do we know the model isn’t just capturing domain-wide patterns (like “cold” usually co-occurring with “cough”) rather than genuine memorization at the individual level? Is there an experiment showing it’s not merely picking up generic associations from the entire dataset? That part still confuses me when considering a single datapoint’s data.

**Relation To Broader Scientific Literature:**

Building on membership inference (Shokri et al.) and memorization (Carlini et al.), the paper generalizes data leakage to weaker assumptions, showing that even scattered fragments can reveal private details.

**Theoretical Claims:**

Yes

---

> ### Author Rebuttal · Authors · 2025-04-01
>
> > **Classifier is considered as the baseline for experiments. Are there any other methods in the literature that you can make the comparison with? / The paper should compare its attack success rate against existing MIA or extraction attacks on the same dataset/model.**
>
> This is an excellent question / comment. To our knowledge, no directly comparable baselines operate under the same weak adversarial assumption as PIFI. Traditional extraction and membership inference attacks require full, ordered samples and much more information (e.g., around 200 tokens), whereas PIFI only assumes access to a small set of unordered fragments (approximately 20). For this reason, we include a classifier -- leveraging ground truth labels -- as a strong, data-aware baseline for performance, against which we compare our data-blind methods (LR-Attack and PRISM). Although we reference likelihood ratio–based methods (e.g., Carlini et al. 2022a) for context, those approaches target a fundamentally different threat model. For instance, Carlini et al. report a 1.4% TPR at 0.1% FPR using 256 shadow models.
>
> While their success rates are in a similar range, direct comparisons are challenging due to significantly differing assumptions. Our goal is to demonstrate that successful extraction is possible even under much weaker assumptions, rather than to claim that PIFI is inherently stronger. We agree that further contextualization alongside prior MIA work would be valuable and will expand on this discussion in the revised paper.
>
> > **(Related question from Reviewer zpv2) Why do we observe that legal setting attacks are more challenging?**
>
> We’d be happy to expand on the legal results in the revised paper, drawing from Appendix B.3 (T6). In the legal summarization task, target fragments often include more common language, including crimes or legal terms that are likely present in general pre-training data. This dilutes fragment-specific signals compared to the medical setting, where terms like “daunorubicin” are more domain-specific and comparatively rare. As a result, attack performance is lower in the legal domain, though still above chance. Interestingly, LR-Attack performs best here, likely due to its heightened sensitivity to rare fragments, which helps in a setting where most targets are otherwise common.
>
> > **But how do we know the model isn’t just capturing domain-wide patterns (like “cold” usually co-occurring with “cough”) rather than genuine memorization at the individual level? / Clarify whether the proposed algorithm leverages the frequency of different conditions?**
>
> These are good questions. Our experiments indicate that the attack isn’t simply leveraging generic domain-wide associations but is also capturing signals specific to individual samples. For example, PRISM’s performance nearly matches that of a data-aware classifier with ground-truth labels. If the model were only exploiting common co-occurrence patterns (e.g., “cold” with “cough”), then the target and shadow models would yield very similar probabilities, and the likelihood ratio would not offer the discriminative power we observe. Moreover, our memorization tests (see Appendix Section E, Table 6) further confirm that fine-tuning leads to the memorization of individual samples.
>
> At the same time, our approach does take advantage of statistical co-occurrence: if an adversary knows that a record contains a fragment like “hypertension,” and the fine-tuned model was trained on data where “hypertension” frequently co-occurs with “osteoporosis,” the model assigns a higher probability to that associated fragment. LR-Attack captures this by comparing the target and shadow model probabilities, while PRISM refines the inference by incorporating a general “world” probability to discount associations that are common across the domain. This combination of mechanisms ensures that our attack can distinguish genuine individual-level memorization from mere generic co-occurrence patterns.
>
> > **(Related question from Reviewer zpv2) In Sec. 4.2, how unique is the given fragment s?**
>
> In the medical setting, we targeted 4,302 fragments, 1,034 of which were unique. Common fragments included "pain" (124), "fever" (69), "shortness of breath" (55), "diarrhea" (47), and "chest pain" (44). However, 75% of fragments appeared fewer than five times, and 47% were targeted only once, indicating that most were highly specific. For instance, single-targeted examples include "Vincristine," "colitis," "daunorubicin," "Naprosyn," "Xalatan," and "lumbar spinal stenosis."

---

### Official Review · Reviewer_wWba · 2025-03-13

**Overall Recommendation:** 2

**Summary:**

This paper introduces a new threat model for extracting sensitive data from fine-tuned LLMs using only partial, unordered fragments. It proposes two data-blind attacks: a Likelihood Ratio Attack and PRISM, which refines inference using an external prior. Experiments in medical and legal domains show these attacks effectively extract private information, rivaling data-aware classifiers.

**Claims And Evidence:**

The paper's claims are generally supported by experimental results and theoretical justifications, demonstrating that fine-tuned LLMs are vulnerable to fragment-based inference attacks. However, some aspects need further validation: (1) PRISM's effectiveness relies on strong assumptions about likelihood ratios; (2) attack robustness against different fragment types is underexplored; (3) generalization beyond medical and legal datasets is limited.

**Essential References Not Discussed:**

The paper discusses relevant prior work on **membership inference and memorization attacks**, but it could benefit from citing additional studies on **fine-tuning vulnerabilities, differential privacy, and fragment-based adversarial attacks**. Below are some references that strengthen the discussion:

1. **Carlini et al. (2022a)** – *Membership inference attacks from first principles* (IEEE S&P 2022)

2. **Nasr et al. (2023)** – *Scalable extraction of training data from (production) language models* (arXiv 2023)

3. **Shokri et al. (2016)** – *Membership inference attacks against machine learning models* (IEEE S&P 2017)
   - One of the **earliest works on membership inference**, forming the basis for **later LLM privacy attacks** like **PIFI**.

**Ethics Expertise Needed:**

["Privacy and Security"]

**Experimental Designs Or Analyses:**

The experimental design is mostly sound, with evaluations on fine-tuned LLMs using medical and legal datasets. The attack success rates at low FPRs support the paper’s claims. However, the analysis lacks a deeper examination of different fragment types, fine-tuning strategies, and broader domain applicability. More ablations on attack robustness across diverse data distributions would improve validity.

**Methods And Evaluation Criteria:**

The proposed methods and evaluation criteria are appropriate for assessing fragment-based privacy risks in fine-tuned LLMs, with relevant medical and legal datasets. However, broader evaluation across different fine-tuning strategies and domains would strengthen the analysis.

**Other Comments Or Suggestions:**

The paper would benefit from **more experiments and a broader range of datasets** to further validate the effectiveness and generalizability of the proposed methods.

**Other Strengths And Weaknesses:**

#### **Strengths**
- The paper introduces a **novel threat model (PIFI)** that extends **membership inference** to cases where the adversary has only **unordered, partial fragments** of sensitive data, making it a significant contribution to **LLM privacy research**.
- The proposed **PRISM method** refines inference using an **external prior**, which improves **robustness against false positives** compared to standard likelihood ratio-based attacks.
- The evaluation in **medical and legal domains** highlights **real-world risks** of fine-tuned LLMs, making the findings **practically relevant**.

#### **Weaknesses**
- The paper lacks **a deeper analysis of why certain fragments are more extractable**, making the attack’s effectiveness less predictable across different datasets.
- **PRISM's assumptions about prior distributions** may not always hold in real-world data, potentially limiting its **generalizability**.
- The study does not provide a **detailed discussion on defenses**, such as differential privacy or fine-tuning strategies that could mitigate fragment-based extraction risks.

**Questions For Authors:**

1. **How do you differentiate between high-confidence ID data and OOD data?**
   - Since both can yield high likelihood scores, what mechanisms ensure that the extracted fragments truly belong to the training data rather than general knowledge?

2. **How does fine-tuning strategy impact attack effectiveness?**
   - The paper evaluates both full fine-tuning and LoRA, but how do other adaptation methods (e.g., RLHF, prompt-tuning) influence vulnerability to PIFI attacks?

3. **How does the model’s exposure frequency to training data affect attack performance?**
   - The results suggest that more fine-tuning epochs increase the risk of fragment inference. Have you explored a threshold where this effect stabilizes?

4. **What are the practical implications for deployed LLMs?**
   - Given that the attack applies to fine-tuned models, how should organizations balance fine-tuning benefits with the privacy risks posed by PIFI?

5. **Would adding noise to model outputs significantly reduce attack success?**
   - The paper mentions that prompt noising has limited impact. Have you tested structured noise approaches, such as adversarial perturbations or differential privacy, as defenses?

6. **How robust is the attack against existing defense methods?**
   - There are many established defense techniques against privacy attacks. How does PIFI perform against **differential privacy, knowledge distillation, adversarial training, or gradient masking**? Have you evaluated its robustness under strong defenses?

**Relation To Broader Scientific Literature:**

The paper extends prior work on membership inference and memorization attacks by introducing PIFI, which extracts sensitive data from fine-tuned LLMs using unordered, partial fragments. LR-Attack builds on likelihood ratio-based inference, while PRISM refines it with a prior to reduce false positives. This contributes to LLM privacy research, highlighting new risks in data leakage and adversarial inference.

**Theoretical Claims:**

The theoretical claims are mostly correct, with LR-Attack and PRISM grounded in established statistical methods. However, PRISM's reliance on prior-based adjustment assumes a strong correlation that may not always hold in real-world cases.

---

> ### Author Rebuttal · Authors · 2025-04-01
>
> > **...assumptions about prior distributions may not always hold in real-world...**
>
> We appreciate the comment. PRISM does make an assumption about the usefulness of a prior world model probability in adjusting the likelihood ratio (assumed to be correlated with sample membership), a premise grounded in statistical principles (see Section 4.2) and supported by prior work [e.g., Carlini et al. 2022a]. Additionally, we have empirically validated this assumption through controlled experiments with a fully specified trigram model (Appendix C) and large-scale LLMs in both medical and legal settings (Section 7, Appendix B). In practice, PRISM reduces false positives compared to LR-Attack, showing that even an approximate prior can provide meaningful signal. Moreover, PRISM performs competitively with, and sometimes better than, a data-aware classifier baseline, suggesting its potential to generalize beyond our setup.
>
> > **Missing citations.**
>
> We discuss all three of these key prior works in our paper already. Carlini et al. (2022a) is cited when introducing membership inference and extraction attacks and motivating our LR-Attack. Nasr et al. (2023) is referenced in our review of scalable extraction methods, supporting our focus on fragment-based extraction in fine-tuned models. Shokri et al. (2016) is acknowledged as foundational in membership inference and helps contextualize our extension of prior threat models. We’d be happy to highlight these works more prominently in the revision.
>
> > **...deeper analysis of why certain fragments are more extractable...**
>
> This is a great question. As noted in our response to Reviewer zpv2, for the medical setting, we targeted 1,034 unique fragments. 47% of those fragments occurred only once. Below, we provide results for the subset of fragments that occur only once, compared to the subset that occurred multiple times:
>
> **Results on target fragments  occurring only once:**
> |Method|TPR@2%FPR|TPR@5%FPR|ROCAUC|
> |--|--|--|--|
> |Classifier|10.8%|13.7%|0.65|
> |LR-Attack|17.5%|34.4%|0.77|
> |PRISM|2.8%|4.2%|0.57|
>
> **Results on target fragments occurring multiple times:**
> |Method|TPR@2%FPR|TPR@5%FPR|ROCAUC|
> |--|--|--|--|
> |Classifier|7.1%|17.0%|0.74|
> |LR-Attack|2.5%|5.3%|0.61|
> |PRISM|5.6%|13.5%|0.73|
>
> This result highlights how incorporating the world_model prob with PRISM significantly improves the sensitivity to more common medical terms, whereas the LR-Attack is very sensitive to rare fragments. We’ll make sure to highlight this result in the revised version of our paper, and add a more in-depth discussion.
>
> > **How does PIFI perform against differential privacy, etc.?**
>
> Thank you, this is an excellent question. **Please see our response to Reviewer m6fY, where we report results on our attack under differential privacy.**
>
> > **How to differentiate between ID and OOD data?**
>
> The LR-Attack could be susceptible to ID vs. OOD data, as you suggest. This motivated the PRISM approach, which uses the world model probability as a prior to adjust the likelihood ratio (target versus shadow), discounting fragments that are common in general knowledge. In principle (see Appendix C for the Synthetic data results, validating this heuristic), PRISM ensures that only fragments with an unusually high likelihood -- beyond what is expected from generic associations -- are flagged as memorized from the training data.
>
> > **How does fine-tuning strategy impact effectiveness?**
>
> Our paper explores this along two dimensions: number of fine-tuning epochs and fine-tuning strategy (full fine-tuning vs. LoRA). We find that more epochs increase TPR at fixed low FPRs, showing that repeated exposure heightens memorization and vulnerability (Section 7.2). While LoRA models are less vulnerable than fully fine-tuned ones, they still leak information, indicating that parameter-efficient methods reduce but don't eliminate privacy risks (Section 7.3). We’re happy to emphasize these findings more in the revision.
>
> > **How does exposure affect attack performance?**
>
> Our experiments show that increasing the number of fine-tuning epochs consistently increases the risk of fragment inference, as demonstrated in Section 7.2. We have observed that attack performance improves continuously with additional epochs. We will add a more in depth analysis to the revised version of the paper, where we step across epochs in greater granularity.
>
> > **Practical implications for deployed LLMs?**
>
> Our work shows that fine-tuning increases memorization risks, even when only partial fragments are exposed. Organizations must weigh the performance gains of fine-tuning against the risk of fragment leakage. Mitigations include using privacy-preserving techniques (e.g., differential privacy), restricting model access, and monitoring for leaks. Ultimately, domain experts should guide these privacy-performance trade-offs through informed risk assessments.

---

> > ### Comment · Reviewer_wWba · 2025-04-06
> >
> > Thank you for the detailed rebuttal. While I appreciate the authors’ empirical efforts and clarifications regarding PRISM and its underlying assumptions, I still find the evaluation of defense methods lacking in depth, particularly in relation to privacy-preserving mechanisms such as differential privacy.
> >
> > Although the authors mention differential privacy briefly in the rebuttal (referring to Reviewer m6fY), there is insufficient analysis on how PRISM performs under strong privacy guarantees or how its assumptions hold when differential privacy is actively applied. Since privacy leakage is central to the paper's motivation, I believe that a more thorough and systematic evaluation in this regard is essential.
> >
> > As such, due to the limited treatment of this important aspect, I will maintain my original score.

---

> > > ### Author Response · Authors · 2025-04-06
> > >
> > > Thank you for your thoughtful feedback. We understand and appreciate your concern regarding our evaluation of defense methods -- specifically, the interplay between our proposed PIFI framework and differential privacy (DP) guarantees. **Due to space constraints in our rebuttal, we were unable to include the full range of our DP experiments (which tested ε values of 0.3, 1, 3, 9, and 27 on the Llama 3.2 3B model).** We will include these extensive results in the revised paper, as noted previously in our rebuttal.
> > >
> > > Furthermore, though we agree that understanding the impact of DP on our attacks is important for settings where developers are mindful of the possibility of such attacks, **we also note that fine-tuning LLMs using DP frameworks is far from the norm in practice, in part because of the significant technical burden of training recent high-performing open LLMs with large batch sizes on a single GPU with a large amount of VRAM** (80GB is often needed for fine-tuning with sufficient batch sizes for 8B-parameter models, even after using techniques like qLoRA to reduce memory requirements). This remains a central issue for DP-fine-tuning, as it necessitates large capital expenditures that many organizations are not willing to commit. It is even a problem that we, as researchers who study these threat models and defenses, have had to contend with in extending our experiments to include models trained under DP.
> > >
> > > Thus, while important, adequate resources for effective DP fine-tuning remain difficult to access for many developers and organizations, leaving many fine-tuned LLMs vulnerable to a threat model such as PIFI. This is to say nothing about the sometimes significant utility trade-offs that scare away many practitioners who fear that using DP will harm their outputs. For an excellent survey that touches on many of these challenges, see Miranda et al. 2024 (https://arxiv.org/pdf/2408.05212). **We believe this further motivates presenting PIFI, as both a real attack and also a cautionary tale to increase the uptake of DP when fine-tuning LLMs.**
> > >
> > > Finally, **we’ll note that many prior memorization/extraction attack papers have not incorporated differentially private training or fine-tuning at all in their experiments** -- for example, Staab et al. (2024 ICLR, https://openreview.net/pdf?id=kmn0BhQk7p), Schwinn et al. (2024 Neurips, https://arxiv.org/abs/2402.09063), and Yu et al. (2023 ICML, https://proceedings.mlr.press/v202/yu23c/yu23c.pdf). Additionally, in works that *do* include results under differentially private training, such as Fu et al. (2024 Neurips, https://openreview.net/pdf?id=PAWQvrForJ) and Lukas et al. (2023 S&P, https://arxiv.org/pdf/2302.00539), they include only a brief section on differential privacy with representative results.
> > >
> > > We adopt a similar perspective to the authors of these papers: **even if DP fine-tuning does reduce the vulnerability of LLMs to our attacks (which it does, and we agree that this is important to understand), that does not diminish the practical importance of the PIFI framework and attacks like PRISM for the vast majority of fine-tuned LLMs,** which do not fine-tune under DP, most commonly due to cost considerations and the lack of awareness of the potential benefits of doing so / fear about the utility cost of introducing noise during fine-tuning. Thus, our revised paper will present in-depth DP results for the Llama 3.2 3B model under PIFI, while deferring a more comprehensive comparison for future work, given the broad scope of the paper already (introducing a new threat model + proposing methods + in-depth evaluations and ablations).
> > >
> > > Even though it appears unlikely that the DP results we have added during the rebuttal will change your mind, **we appreciate your perspective and hope that the above discussion coupled with our inclusion of the additional DP experiments in the revised paper will sufficiently address your concerns.** Thank you again for engaging with us during the rebuttal phase.

---

### Official Review · Reviewer_zpv2 · 2025-03-13

**Overall Recommendation:** 3

**Summary:**

The increasing development of large language models (LLMs) has resulted in different explorations of their trustworthiness properties. Amongst them, privacy is one of the key concerns, where prior research has shown that LLMs are prone to leaking sensitive training data through memorization and membership inference attacks. One of the main bottlenecks with existing works is that adversarial attackers assume complete access to the training samples or some ordered prefixes. In this work, the authors explore a novel direction for testing the vulnerability of LLMs when adversarial attackers have access to only partial and unordered sample information. In particular, the authors propose LR-Attack and PRISM threat models and show that fine-tuned LLMs are susceptible to fragment-specific extraction attacks. Using small datasets from medical and legal domains, the authors show the effectiveness of the proposed attacks. While the questions raised by the authors are interesting, the paper lacks rigorous evaluation and misses key insights (please refer to the sections below for more details). Overall, the paper reads well and the authors have explored an interesting direction!

## update after rebuttal

Thank you for your detailed rebuttal response and for addressing my concerns. I will stick with my weak accept rating as it's unclear why smaller models will overfit such a dataset and the proposed LR-Attack and PRISM will outperform the classifier baseline.

**Claims And Evidence:**

The claims are supported by experimental results but they are not conclusive and haven't been thoroughly evaluated. Please see below for some open questions and refer to the "Methods and Evaluation Criteria" for more details.

i) In Sec. 4.2, how unique is the given fragment $s$?

ii) The classifier baseline states that it should always achieve the best score as it uses a ground-truth label, but still, in Sec. 7, we observe that LR-Attack and PRISM outperform the classifier baseline. It would be great if the authors could explain these results further.

iii) How transferrable are these attacks to other open- or closed-source LLMs?

iv) Algorithm 1 describes the use of the decision threshold. Is this threshold different for different samples in a given dataset?

v) How would these attacks perform when tested on safety-tuned models or models fine-tuned with privacy constraints?

vi) Intuitively, it seems that the proposed algorithm is leveraging the frequency of different conditions (in the medical dataset) that occur together, i.e., hypertension and osteoporosis frequently occur together in medical notes. Can the authors clarify this?

**Essential References Not Discussed:**

NA

**Experimental Designs Or Analyses:**

Yes, I read the experimental setup and thoroughly reviewed the analysis.

**Methods And Evaluation Criteria:**

The proposed method and evaluation criteria make sense within the context of understanding whether LLMs are vulnerable to adversaries who only have partial and unordered sample information. However, the evaluation of the proposed methodology is not very strong. For instance,

i) the authors test their proposed attacks using two small datasets (medical summarization and legal setting), which contain 312 and 235 test samples, respectively.

ii) the authors mostly use small language models in the range of 3-8B parameters, which raises the question of the utility of the proposed attack on large language models (like Llama 13b and other models in higher parameter range).

iii) The main results compare LR-Attack and PRISM with a random-guess baseline, where the proposed methods obtain an AUC of ~0.65, on average. Are these results conclusive to showcase the effectiveness of the proposed algorithms?

iv) There are no results to show the robustness of the method against existing defense techniques.

v) The results for the legal settings are interesting, but the authors do not discuss them in detail, e.g., why do we observe that legal setting attacks are more challenging? Similar arguments for other results (like in 7.7), where the authors do not provide any explanation of the obtained results.

vi) The authors propose to use LightGBM for the classifier baseline but do not provide the accuracies obtained by LightGBM on the classification task. Since the experiment section doesn't comprise any existing baselines, it would be beneficial to have a range of highly predictive classifier baselines to show the effectiveness of LR-Attack and PRISM.

vii) In Figures 4-5, why does LR-Attack obtain an AUC lower than random guess?

viii) In Appendix F, the authors mention that they take the mean of the probabilities output by Llama-3.1-8B-Instruct, Mistral-7B-v0.2, and Gemma-2B-IT LLMs. Why not use an ensemble of large open-sourced models as the world model?

**Other Comments Or Suggestions:**

NA

**Other Strengths And Weaknesses:**

The paper presents an interesting perspective in attacking large language models using partial-information fragments.

**Questions For Authors:**

Please refer to the "Claims And Evidence" and "Methods And Evaluation Criteria" for more details.

**Relation To Broader Scientific Literature:**

The paper has a key contribution to the broader scientific literature of memorization and membership inference by proposing threat models that can break LLMs using partial and unordered information.

**Theoretical Claims:**

NA

---

> ### Author Rebuttal · Authors · 2025-04-01
>
> > **In Sec. 4.2, how unique is the given fragment s?**
>
> **Please see our response to Reviewer rGXA, where we give exact details on uniqueness.**
>
> > **Why does LR-Attack and PRISM sometimes outperform classifier baseline?**
>
> This is a great question and we’ll include this discussion in the revised paper — while the classifier baseline benefits from ground-truth labels and can, in theory, exploit signals more effectively, its performance may degrade due to overfitting, especially when fragment distributions are highly variable. In contrast, LR-Attack and PRISM use data-blind decision rules based on probability ratios (and a Bayesian update in PRISM), without learning from labeled data. This makes them more robust to overfitting. As a result, in settings with high distributional variation, these rigid decision rules can sometimes outperform the classifier.
>
> > **Attacks transferable to other open- or closed-source LLMs?**
>
> We evaluated our attack on a range of open models and found it consistently effective across architectures and parameter scales (see Appendix B1, Figure 7). The transferability to closed models, however, is currently limited by the need for echo functionality on logprobs — previously available in some APIs (e.g., OpenAI’s) but now removed due to security concerns. Future work could explore alternatives, such as methods proposed by Finlayson et al. (2024) to obtain logprobs from closed models.
>
> > **Is this threshold in algorithm 1 different for different samples in a given dataset?**
>
> Good question, we’ll clarify this in the revised paper. No, the threshold is fixed for each strategy based on the desired sensitivity, similar to prior membership inference attacks [Carlini et al. 2022a; Duan et al. 2024]. For instance, on a converged Llama 8B model, the PRISM threshold is 0.081, and the LR-Attack threshold is 1.78 when targeting a 2% FPR.
>
> > **Test on models fine-tuned with privacy constraints?**
>
> Thank you for the excellent suggestion. **Please see our response to Reviewer m6fY, where we evaluate our attack under a differential privacy.**
>
> > **Clarify whether the proposed algorithm leverages the frequency of different conditions?**
>
> Thank you for this observation. **Please see our response to Reviewer rGXA, where we discuss frequency / co-occurrence**
>
> > **…test using small datasets…**
>
> While the test datasets are relatively small (312 samples for medical and 235 for legal), our attack evaluated over 4,302 target fragments, providing strong evidence for the feasibility of the PIFI threat model. We plan to expand our evaluation to larger datasets in future work to further validate these results.
>
> > **…what is utility of the proposed attack on larger LLMs?**
>
> Thank you for the suggestion; we have evaluated our attack on a 70B-parameter Llama-3.3 model fine-tuned for 10 epochs, which is the largest model we could finetune given our compute constraints. The results below show that the attack remains effective even at this larger scale:
> |Method|TPR@2%FPR|TPR@5%FPR|ROCAUC|
> |-|-|-|-|
> |Classifier|4.9%|11.4%|0.67|
> |LR-Attack|5.2%|10.2%|0.64|
> |PRISM|4.2%|11.3%|0.65|
>
> We will include this result in our revisions.
>
> > **Why do we observe that legal setting attacks are more challenging?**
>
> **Please see our response to Reviewer rGXA, where we discuss the legal setting in detail.**
>
> > **…do not provide LightGBM accuracies on the classification task + range of performant classifiers...**
>
> We'd be happy to report accuracies for the LightGBM classifier baseline + other classifier baselines (e.g., XGBoost, FT-Transformer etc.). For example, the Llama-3 8B model attack had accuracy (0.77) and F1 score (0.25); however, we believe in our focus on ROC AUC and TPR (which we do report on the classifier baseline) as it reflects the utility of our LR-Attack and PRISM methods in low-FPR regimes (consistent with [Carlini et al. 2022a]).
>
> > **…Figures 4-5, LR-Attack obtains an AUC lower than random guess?**
>
> This typically means the scoring function is inversely correlated with true membership, and happens when assumptions about target vs. shadow model behavior break down, such as when both assign similar probabilities or the fragment is very common, skewing the ratio. This is one motivation for PRISM.
>
> > **…ensemble of large open-sourced models as the world model?**
>
> Great suggestion. We have now experimented with using DeepSeek-v3/r1 world probabilities (averaged), and found that this higher-quality model did improve performance. These results suggest that adding more high-quality open-source models to the world model ensemble could further enhance our approach.
>
> Below is the performance of the Llama 3 8B model with DeepSeek world probabilities (10 epochs):
> |Method|TPR@2%FPR|TPR@5%FPR|ROCAUC|
> |--|--|--|--|
> |Classifier|7.0%|13.4%|0.69|
> |LR-Att.|5.3%|10.6%|0.64|
> |PRISM|5.2%|11.6%|0.7|

---

### Official Review · Reviewer_m6fY · 2025-03-16

**Overall Recommendation:** 4

**Summary:**

This paper introduces a new privacy threat model, Partial-Information Fragment Inference (PIFI), which examines how adversaries can extract sensitive information from LLMs using only small, unordered text fragments rather than full training samples. Unlike traditional memorization or membership inference attacks, which assume strong adversarial access (e.g., full samples or ordered prefixes), PIFI explores a weaker but more realistic scenario where attackers infer hidden details from publicly available fragments. The authors propose two data-blind attack methods: Likelihood Ratio Attack (LR-Attack) and PRISM (Posterior-Refined Inference for Subset Membership), which leverage statistical techniques to infer missing private fragments. Their empirical evaluation on fine-tuned LLMs in medical settings shows that these attacks can successfully extract sensitive information with a non-trivial success rate, even with limited adversarial knowledge. The study highlights vulnerabilities in fine-tuned LLMs and suggests that existing defenses focusing on memorization or membership inference are insufficient. The authors emphasize the need for new mitigation strategies before deploying LLMs in sensitive domains like healthcare or law.

**Claims And Evidence:**

The claims in the submission are largely supported by empirical evidence, but some areas may require further clarification or stronger justification. Here’s an analysis of key claims:

Claim: Adversaries can extract sensitive information from LLMs using unordered fragments.

Potential issue: The paper presents two attack methods (LR-Attack and PRISM) and evaluates them on fine-tuned LLMs in a medical setting. The results show non-trivial success rates, supporting the claim that fragment-based inference is feasible. The generalizability of this finding beyond the tested fine-tuned models is unclear. The study should discuss whether the attacks work on non-fine-tuned, general-purpose LLMs.

Claim: PIFI is a more realistic threat model than prior work on memorization and membership inference.
The motivation for PIFI is reasonable—real-world adversaries may only have access to partial data.
Potential issue:
The paper should compare its attack success rate against existing membership inference or extraction attacks on the same dataset/model to substantiate its claim of PIFI being a stronger or more practical threat.
Claim: The proposed attacks (LR-Attack and PRISM) are effective.

**Essential References Not Discussed:**

NA

**Experimental Designs Or Analyses:**

The experimental design is reasonable. The proposed attack methods are well-motivated and align with plausible LLM privacy vulnerabilities.

**Methods And Evaluation Criteria:**

Measuring fragment reconstruction accuracy and attack success rate is logical.
Potential Issues: It is unclear whether the paper accounts for trivial guesses—does the attack outperform a simple "most likely completion" heuristic?

The claim that existing defenses may not fully prevent PIFI attacks is interesting and relevant.
The paper should explicitly evaluate common privacy-preserving techniques. If tested defenses are ineffective, discussing why they fail would add depth to the analysis.

The PRISM attack is not completely clear to me. A more intuitive explanation would help.

**Other Comments Or Suggestions:**

Better clarity in writing will help specially in technical sections

**Other Strengths And Weaknesses:**

See above sections

**Questions For Authors:**

See previous sections.

**Relation To Broader Scientific Literature:**

The key contributions of the paper build upon prior work in LLM privacy risks, membership inference, and data extraction attacks. Specifically, it extends findings in the following areas:

- Data Memorization in LLMs and Membership inference
- Attacks based on statistical inference
- Prompt-Based Extraction Attacks:

**Theoretical Claims:**

NA

---

> ### Author Rebuttal · Authors · 2025-04-01
>
> > **…discuss whether the attacks work on general-purpose LLMs**
>
> Thank you for raising this important point. Our work specifically focuses on sensitive domains (e.g., hospitals or legal) where models are adapted on private data, and fine-tuning is often necessary because organizations cannot or do not wish to train a full-scale foundational model on sensitive data. Consequently, the vulnerabilities we expose are most pertinent to these fine-tuned models. Additionally, prior work (e.g., [Duan et al. 2024]) shows that membership inference attacks on foundation models are significantly more challenging. as such they’re trained on broad datasets and memorize less. Additionally, our method depends on building a shadow model, a step that is often impractical for large, non-fine-tuned models due to their scale and complexity. We also note, though, that our fine-tuned models remain performant on general-purpose benchmarks like MMLU, suggesting they can be used in manner similar to other general-purpose models, as discussed further in our response to reviewer wf5m.
>
> > **…compare its attack success rate against existing MIA or extraction attacks...**
>
> Thank you for the comment; some other reviewers brought up similar points. **Please see our response to Reviewer rGXA, where we discuss MIA performance relative to our PIFI threat model.**
>
> > **…unclear whether the paper accounts for trivial guesses… does attack outperform a "most likely completion" heuristic?**
>
> Thank you for the suggestion. To clarify, by “most likely completion” do you mean a baseline that considers a greedy or Monte Carlo generation of the next token? We’d happy to include another baseline, please clarify and we will add it in the revised paper.
>
> > **…explicitly evaluate common privacy-preserving techniques.**
>
> This is a great suggestion. **We have added experiments evaluating differentially private (DP) finetuning on the Llama 3.2 3B Instruct model using sample-level DP (via the dp-transformers package with ε = [0.3, 1, 3, 9, 27] over 10 epochs).** We include only the ε = 3 results here due to space constraints, but will add full results in the revised paper. This table shows task performance for the base model, the DP fine-tuned (FT) model, and the non-private FT model:
>
> |Metric|BaseModel|DPFTModel($\epsilon=3.0$)|Non-Priv.FTModel|
> |--|--:|--:|--:|
> |ROUGE-Lsum|0.0963|0.0969|0.1004|
> |BERTScoreF1|0.7140|0.7186|0.7299|
>
> DP FT slightly improves performance over the base model but does not reach the performance of non-private FT, which aligns with expectations [Lukas et al. 2023].
>
> We further evaluated the PIFI threat model and attacks on the DP FT LLM:
>
> |Method|TPR@2%FPR|TPR@5%FPR|ROCAUC|
> |--|--:|--:|--:|
> |Classifier|4.4%|10.0%|0.64|
> |LR-Attack|0.9%|2.4%|0.51|
> |PRISM|4.0%|9.6%|0.54|
>
> Here, DP fine-tuning reduces the success of the LR attack (0.9% TPR at 2% FPR), indicating the DP mechanism protects against this attack. However, both the classifier baseline and PRISM still achieve roughly twice the TPR at fixed low FPRs. We consider a potential explanation:
>
> Sample level DP guarantees for LLM finetuning ensure $\sum_{i=1}^T \ell_i(x_{1:i}) \leq \epsilon$ where $\ell_i(x_{1:i}) = \log \frac{\Pr(x_i \mid x_{1:i-1}, D)}{\Pr(x_i \mid x_{1:i-1}, D')}$ (Yu et al., 2021). However, note that this guarantee of privacy loss can be unevenly distributed: for example, consider a two-token case with $\frac{\Pr(x_1 \mid D)}{\Pr(x_1 \mid D')} = e^\epsilon$ and $\frac{\Pr(x_2 \mid x_1, D)}{\Pr(x_2 \mid x_1, D')} = 1$; the total privacy loss $L(x_1, x_2) = \epsilon + 0 = \epsilon$, satisfying the guarantee, but the entire privacy budget is focused on a single token! In summary, with standard DP finetuning **individual tokens can incur nearly $\epsilon$ loss if others compensate.** Further analysis + other DP approaches would be promising for future work.
>
> > **…PRISM attack is not completely clear to me…more intuitive explanation would help**
>
> Thank you, we’ll include this discussion in the revised paper if helpful. PRISM builds on the standard LR attack by asking: how surprising is it that the target fragment appears, given what we expect in general? LR attack compares the target model  (which has seen the sample) and shadow model (which hasn’t) probabilities to detect memorization of a fragment. However, this ratio can be high simply because a fragment is common in a domain. PRISM corrects by incorporating a “world model” that estimates the general likelihood of a fragment. In essence, PRISM performs a Bayesian update — it adjusts LR score with a prior that reflects how likely the fragment is in any sample. If "smoker" frequently co-occurs with "osteoarthritis," a high LR score might not indicate memorization; by using the world model probability, PRISM tempers the score -- if "smoker" is common overall, the adjusted score is lower, reducing FPs. We validate this with a small, controlled synthetic setup (Appendix C), where PRISM outperformed the basic LR attack.

---

### Official Review · Reviewer_wf5M · 2025-03-24

**Overall Recommendation:** 2

**Summary:**

This paper introduces a novel Partial-Information Fragment Inference (PIFI) threat model that examines the potential for sensitive data extraction from LLMs using only unordered, publicly available fragments of information. Two data-blind attack methods are proposed: LR-Attack (likelihood ratio-based) and PRISM (posterior-refined), which aim to infer whether a private fragment was part of an individual’s data used in model fine-tuning. Experiments in medical and legal domains show that even limited adversarial knowledge enables meaningful privacy breaches, highlighting new privacy vulnerabilities in LLMs.

**Claims And Evidence:**

The key claim is that LLMs fine-tuned on sensitive data are vulnerable to fragment-level privacy attacks under weak assumptions. This is supported by empirical results showing non-trivial TPR at low FPR in both medical and legal domains using the proposed methods. However, the results, while statistically significant, are modest in absolute terms (e.g., ~10% TPR @ 2–5% FPR), raising questions about practical impact. Furthermore, while the authors argue that such attacks are plausible in real-world settings, the practicality and prevalence of the assumed attacker capabilities remain somewhat speculative.

**Essential References Not Discussed:**

No.

**Experimental Designs Or Analyses:**

The experiments are well-structured and thorough, including multiple models (e.g., LLaMA, Qwen, Mistral), LoRA variants, and varying fine-tuning depths. The ROC curve analysis is detailed, and sensitivity studies (e.g., fragment noise, model scale) are insightful. However, the practical severity of attacks remains underexplored. For example, how often would such fragment sets occur in the wild, or how adversaries might obtain them reliably.

**Methods And Evaluation Criteria:**

The paper uses realistic datasets (e.g., MTS-Dialog for medical records), a clear delineation of attack models, and fair baselines (e.g., Classifier as a data-aware oracle). Evaluation metrics such as TPR@FPR and AUC are appropriate, and the inclusion of shadow and world models demonstrates a careful design. The novel aspect lies in relaxing the assumption of access to complete training samples, aligning PIFI more closely with real-world scenarios where attackers may only have limited context.

**Other Comments Or Suggestions:**

- Consider adding utility evaluations (e.g., GLUE or MMLU) of LLMs before and after fine-tuning to evaluate whether models retain general abilities?

- Clarify the practical steps by which an attacker might construct fragment sets S in open domains.

- Add qualitative examples of successful and failed attacks to aid interpretability.

**Other Strengths And Weaknesses:**

Strengths:

- Thorough experiments across domains and model types.

- Proposed methods are data-blind, making the attack scenario plausible.

Weaknesses:

- Practicality of the threat model is questionable in real-world adversarial scenarios.

- Utility evaluation of fine-tuned models is missing. Do LLMs still generalize after fine-tuning, or are they just memorizing?

- Attacks yield modest absolute TPR values, which may limit their practical threat significance.

**Questions For Authors:**

- Have you evaluated model generalization on standard NLP benchmarks pre- and post-finetuning to evaluate levels of overfitting? E.g.,  GLUE or MMLU.

- How plausible is it for an attacker to reliably obtain the fragment sets you assume (e.g., 10–30 specific keywords)?

- Could the attack work without knowing the exact fragments from the sample, i.e., based on approximate knowledge?

- How sensitive are the results to the Prompt(S) template used?

**Relation To Broader Scientific Literature:**

The paper is well-situated in the ML privacy literature, extending work on memorization and membership inference.

**Theoretical Claims:**

No formal proofs are presented.

---

> ### Author Rebuttal · Authors · 2025-04-01
>
> > **Concerns about practical impacts, given the “modest” results.**
>
> We appreciate the concern about practical impact. **While a 10% TPR at 2-5% FPR may seem modest, it can still pose a significant privacy threat (e.g., it equates to an attacker being correct 4 out of 5 times), especially scaled to thousands of individuals.** Moreover, membership inference attacks also face challenges in achieving high TPRs, yet are considered to be a significant privacy concern (see e.g., Duan et al. 2024)
>
> > **The practicality and prevalence of the assumed attacker capabilities remains speculative. How plausible is it for an attacker to reliably obtain the fragment sets you assume (e.g., 10–30 specific keywords)?**
>
> We agree that the threat model’s practical impact depends on how feasible it is for an attacker to gather necessary fragments. In domains like healthcare, such fragment data is increasingly accessible. **A 2020 study [Seh et al., 2020] found that healthcare data breaches between 2005 and 2019 affected over 249 million individuals.** Additionally, self-health data collection (e.g., Lupton, 2016) and sharing on social media make such fragments more easily available. Given something as simple as a public record or leaked insurance info, combined with a reliable set of fragments derived from otherwise benign medical data shared on social media, an attacker could construct an effective set S of tokens to enable targeted attacks, as demonstrated by our PIFI threat model. We will elaborate on this in the revised paper.
>
> > **Generalization results on standard NLP benchmarks (MMLU) pre- and post-finetuning**
>
> Thank you for the suggestion. **We ran preliminary utility evaluations (MMLU). The Llama-3-8B model fine-tuned for 10 epochs reached an average MMLU score of 0.565, compared to 0.487 after one epoch and 0.413 for the baseline model.** In medical areas (e.g., clinical knowledge, medical genetics), fine-tuning improved performance over a non-fine-tuned baseline, suggesting the model remains capable and general.  Note that further fine-tuned models can outperform generalist models on some benchmarks due to “better behavior” (they naturally follow a stricter format), which DeepEval docs specifically mention as a limitation of MMLU.
>
> | MMLU Category | No Fine-Tune (Zero-Shot) | 1-Epoch Fine-Tune (Zero-Shot) | 10-Epoch Fine-Tune (Zero-Shot) |
> |--|--:|--:|--:|
> | Clinical Knowledge | 0.41 | 0.57 | 0.63 |
> | Professional Medicine | 0.57 | 0.64 | 0.69 |
> | Abstract Algebra | 0.27   | 0.15  | 0.25 |
> |…|…|…|…|
>
> We will add exhaustive MMLU results to our revised paper.
>
> > **Add qualitative examples of successful and failed attacks to aid interpretability.**
>
> Here are several examples of successful and failed attacks under PIFI from our data. We will update the paper with several of these in accordance with your recommendation.
>
> 1. An attack with the fragments “anxiety disorder, arthritis, Morton's neuromas, migraines” infers that an individual has hypothyroidism.
> 2. An attack with the fragments “shortness of breath, Coumadin, lightheadedness, chest pain, Cardizem, pain, vertigo” infers that an individual has atrial fibrillation.
> 3. An attack with the fragments “toxicity, breast cancer, Ixempra, tumor, neuropathy, Avastin, cancer, Faslodex, Zometa, ixabepilone” infers that an individual takes Aromasin.
>
> Failed Inferences:
> 1. An attack with the fragments “cramping, trauma, pain, shock, numbness” fails to infer that an individual takes Naprosyn.
> 2. An attack with the fragments “diabetes, redness, swelling, itchiness, pain, skin infection” fails to infer that an individual also has cellulitis.
>
> False Positives:
> 1. An attack with the fragments “myelofibrosis, swelling, diarrhea, hydroxyurea, J A K, steroids, polycythemia vera, lenalidomide, smoking, smoke, pain, numbness” incorrectly infers that an individual takes Prednisone.
> 2. An attack with the fragments “lung disease, shortness of breath, pneumonia, oxygen” incorrectly infers that an individual has COPD.
>
> > **Could the attack work without knowing the exact fragments from the sample, i.e., based on approximate knowledge?**
>
> Yes, the attack remains effective when the attacker’s knowledge is approximate. **In our ablation studies (see Appendix, Figure 9), we show that replacing 25-75% of true fragments with random unrelated ones leads to only modest performance drops.** This indicates the method is robust even when the adversary’s information is imperfect.
>
> > **How sensitive are the results to the prompt(s) template used?**
>
> We experimented with various prompt templates (e.g., “Consider a patient’s medical summary includes…”; “Suppose a patient’s medical summary includes conditions such”; etc.) and found that attack performance did not change substantially. While format influences raw output probabilities slightly, the overall ranking of candidate fragments (and hence the attack’s effectiveness) remains consistent. We will add a sensitivity analysis in the revised paper.

---

### Decision · Program_Chairs · 2025-05-01

**Decision:**

Accept (poster)

**Comment:**

This paper proposes a new threat model termed PIFI --- where an adversary only has partial knowledge of the training data to be inferred and only cares about membership signal on fragments rather than an entire sample.

Reviewers found both many strengths and weaknesses to this work. This paper proposes an interesting threat model and mostly thorough experiments (several models + datasets, though limited in scale on both). In the revision, the authors also strengthened their empirical evaluation with further utility evaluation of the fine-tuned models. The attacks presented, though heavily inspired by prior work, are of interest as they present strong attacks for this new proposed threat model.

However, reviewers also had several issues that were not fully addressed. Notably, there is lacking comparison with existing attacks (membership inference and data extraction) and lacking strong baselines. Indeed, proposed upper bounds for attack performance empirically are shown to be surpassed by the proposed attacks without convincing explanation as for why. There are also several aspects related to the novelty of the fragment threat model that are underexplored.